# Stable water splitting using photoelectrodes with a cryogelated overlayer

Byungjun Kang [1,4], Jeiwan Tan[2,3,4], Kyungmin Kim [2], Donyoung Kang [1], Hyungsoo Lee [2], Sunihl Ma [2], Young Sun Park [2], Juwon Yun[2], Soobin Lee [2], Chan Uk Lee [2], Gyumin Jang [2], Jeongyoub Lee [2], Jooho Moon [2] ✉ & Hyungsuk Lee [1] ✉

Hydrogen production techniques based on solar-water splitting have emerged as carbon-free energy systems. Many researchers have developed highly efficient thin-film photoelectrochemical (PEC) devices made of low-cost and earth-abundant materials. However, solar water splitting systems suffer from short lifetimes due to catalyst instability that is attributed to both chemical dissolution and mechanical stress produced by hydrogen bubbles. A recent study found that the nanoporous hydrogel could prevent the structural degradation of the PEC devices. In this study, we investigate the protection mechanism of the hydrogel-based overlayer by engineering its porous structure using the cryogelation technique. Tests for cryogel overlayers with varied pore structures, such as disconnected micropores, interconnected micropores, and surface macropores, reveal that the hydrogen gas trapped in the cryogel protector reduce shear stress at the catalyst surface by providing bubble nucleation sites. The cryogelated overlayer effectively preserves the uniformly distributed platinum catalyst particles on the device surface for over 200 h. Our finding can help establish semi-permanent photoelectrochemical devices to realize a carbon-free society.

Solar-hydrogen production using water-splitting photoelectrodes is a promising technology to realize a carbon-free society[1]. For the practical use, it is required to develop photoelectrodes with high solar-to-hydrogen conversion efficiency, low fabrication cost, and long lifetime. State-of-the-art photoelectrodes with high solar-to-hydrogen efficiency generally consist of high-cost and rare materials, such as III–V group light absorbing semiconductors, that limit the practical application of photoelectrochemical (PEC) systems[2]. Alternatively, researchers have intensively investigated low-cost and earth-abundant light-absorbing semiconductors, such as $TiO_2$[3], $Fe_2O_3$[4], $BiVO_4$[5], $Cu_2O$[6], $Cu_2ZnSnS_4$[7], and $Sb_2Se_3$[8], which are typically fabricated into thin-film photoelectrodes. Advanced interface engineering of thin-film photoelectrodes, including heterojunction formation and surface

decoration of the co-catalyst, effectively improves the photovoltage and photocurrent of devices[9]. However, the lifetime of thin-film photoelectrodes remains insufficient for commercial use because they suffer from severe photocorrosion, which competes with the water-splitting reaction at the photoelectrode/electrolyte interface. Use of co-catalysts such as Pt on the photoelectrode surface can drive photogenerated charges towards the favored water-splitting reaction by reducing the redox overpotential instead of the photocorrosion reaction. Conformal coating of protection layer (e.g., $TiO_2$) typically extends the device lifetime by physically separating the semiconductor from the corrosive electrolyte. However, we observed that the sluggish kinetics of surface electrochemical reaction (e.g., HER) could lead to the dissolution of $TiO_2$[10]. Despite uniform $TiO_2$

[1]School of Mechanical Engineering, Yonsei University, Seoul 03722, Republic of Korea. [2]Department of Materials Science and Engineering, Yonsei University, Seoul 03722, Republic of Korea. [3]Present address: Chemistry and Nanoscience Center, National Renewable Energy Laboratory, Golden, CO 80401, USA. [4]These authors contributed equally: Byungjun Kang, Jeiwan Tan. ✉e-mail: jmoon@yonsei.ac.kr; hyungsuk@yonsei.ac.kr

deposition on semiconductor, agglomeration and detachment of the co-catalyst from the photoelectrodes can accelerate the reductive dissolution and the photocurrent degradation reducing the lifetime of device.

Various co-catalyst protection strategies have been suggested, mostly focusing on improving the adhesion between the photoelectrode and catalyst[11–14]. For example, a Pt catalyst-encapsulating strategy with a few nanometer-thick solid oxide overlayers extended the lifetime of photoelectrodes with a flat surface of a Si wafer[12]. However, a uniform coating of a few nanometer-thick oxide layers onto thin-film photoelectrodes, which generally have non-flat nanostructured surfaces, is experimentally challenging. This inevitably produces pinholes and cracks over these layers[11,12], which might be influenced by the bubbles produced at the catalyst/overlayer interface during PEC operation. Therefore, it is necessary to develop a versatile protection strategy to increase the lifetimes of PEC devices for practical applications.

We hypothesized that the structural stability of the device could be improved by regulating the mechanical stress applied on the co-catalyst during the gas evolution reaction. Once the hydrogen or oxygen molecules produced at the co-catalysts exceed the critical concentration required to overcome the free-energy barrier of bubble nucleation, they follow the nucleation and growth theory along the device surface until they detach by increasing buoyancy[15]. The surface-growing bubbles apply mechanical force to the device surface at the bubble-device interface[16] (Supplementary Fig. 1), which may cause migration and aggregation of the co-catalysts[17,18]. The surface nanostructuring, coating of aerophobic polymer, or addition of surfactant to the electrolyte solution could reduce the critical size of bubble required for the bubble detachment[19–24]. However, shear stress is still produced on the catalyst surface by bubbles in a periodic manner through repeated cycles of bubble nucleation, expansion, and detachment.

The recent study suggested that the coating of the nanoporous hydrogel on the PEC device could significantly enhance the structural stability of the catalyst and PEC devices[25]. The structural stability of the catalyst layer might be attributed to the immobilization of catalyst particles by the spatial confinement of hydrogel or reduction of shear stress on the catalyst by regulating the bubble dynamics. In this study, we sought to understand the mechanism of how the porous protector improves the structure stability of the device by incorporating overlayer having various types of porous structures. We employ a cryogelation technique to fabricate a microporous hydrogel overlayer because it is advantageous over other techniques such as porogen/leaching and gas-foaming ones in terms of fabrication speed and simplicity[26]. Experiments show that, during the PEC operation, gas bubbles are nucleated and trapped in the cryogel to provide nucleation sites of bubbles for transporting gas molecules to the outer electrolyte solution. Porous structure of the cryogel overlayer contributes to reduce the mechanical shear stress produced by nucleated bubbles at the device surface preserving the surface structure of the Pt/TiO2/Sb2Se3 photocathode even after 210 h operation. Theoretical and numerical analyses reveal how the structural and functional stabilities of the PEC devices are determined by mechanical fracture and softening of the protector occurred by bubbles. The scientific advances in this study can be applied to develop semi-permanent electro- and photoelectro-chemical devices.

## Results

### Fabrication of microporous overlayer by cryogelation
We fabricated an interconnected porous polyacrylamide cryogelated overlayer on top of a Pt/TiO2/Sb2Se3/Au/fluorine-doped tin oxide configured photocathode (Fig. 1a). The sputtered Pt nanoparticles, atomic layer-deposited (ALD) TiO2, and solution-processed nanostructured Sb2Se3 work as a co-catalyst, protection layer, and light

absorber, respectively. The photoelectrons generated in $Sb_2Se_3$ were transported to the Pt catalyst through the $TiO_2$ layer and reacted with hydrogen ions to produce hydrogen gas molecules, which formed gas bubbles. The produced gas bubbles escaped through the pores of the cryogelated overlayer.

Cryogelation of the polyacrylamide hydrogel was conducted in a freezer (−20 °C) (Supplementary Fig. 2)[26]. During the early stage of freezing, isolated ice crystals nucleated at several locations in the pre-gel solution. After a critical freezing duration, the growing ice crystals interconnected[27]. Phase separation of the pre-gel solution into frozen and non-frozen phases occurred during the formation of ice crystals[26]. Reactive entities of the hydrogel were concentrated in the nonfrozen phase between the ice crystals underwent free-radical polymerization[28] to form the polymer network. Thawing of the ice crystals after polymerization produced a microstructured polyacrylamide overlayer. The optical transmittance of the cryogel was higher than 90% for the wavelengths ranging from 400 to 1000 nm (Supplementary Fig. 3). The internal structure of the cryogelated overlayer was visualized by three-dimensional confocal fluorescence imaging of 200 nm fluorescent nanoparticles attached on the polymer (Fig. 1b and Supplementary Fig. 2). The average pore size was approximately 20 μm which is much larger compared to catalyst nanoparticles. Thus, the effect of spatial confinement on the catalyst by the cryogel would be smaller than that by a regular hydrogel with nanopores used in the previous study[25]. The average area fraction of the micropores was estimated to be approximately $59.3 \pm 3.5\%$ for cryogels (Supplementary Fig. 4). The polymeric part of the cryogel overlayer is polyacrylamide hydrogel which is nanoporous, poroelastic, water-permeable, hydrophilic, and aerophobic[25,29,30]. The polyacrylamide hydrogel was tested to be stable under light illumination[31,32].

### Effect of cryogelated overlayer on photoelectrochemical characteristics
To investigate the effect of the microporous cryogel overlayer on the performance and the structural stability of the PEC devices, we performed chronoamperometry at 0 V versus a reversible hydrogen electrode ($V_{RHE}$) under 1-sun illumination on various $Sb_2Se_3$ photocathodes covered with cryogel overlayers having thicknesses of 0, 100, 200, 400, 800, and 1200 μm, denoted as *no cryogel*, *CG-100 μm*, *CG-200 μm*, *CG-400 μm*, *CG-800 μm*, and *CG-1200 μm*, respectively. An initial value of photocurrent density ($J_o$) was not affected by the cryogel coating significantly (Supplementary Fig. 5). As shown in Fig. 1c, *no cryogel* showed an $J_o$ of 16 mA cm$^{-2}$ and an irregular fluctuation in the photocurrent density ($J_{ph}$) for 1 h. For *CG-400 μm*, $J_{ph}$ decreased from 16 to 7 mA cm$^{-2}$ during the first 0.1 h, followed by a partial recovery to 11 mA cm$^{-2}$. Interestingly, after the $J_{ph}$ recovery, the $J_{ph}$ was maintained with small and regular fluctuations. *CG-800 μm* and *CG-1200 μm* exhibited a large initial $J_{ph}$ drop and recovery within 0.5 h, followed by a slight $J_{ph}$ fluctuation. The initial parabolic decreasing profile during the initial $J_{ph}$ drop implies that the produced bubble continuously grows inside the cryogel, which would prevent the scattering of incoming light and reduce the mass transfer of electrolytes, thereby temporarily reducing the $J_{ph}$ of devices. $J_{ph}$ was instantly recovered as the trapped bubble escaped. After this recovery, the $J_{ph}$ of *CG-400 μm*, *CG-800 μm*, and *CG-1200 μm* was stably maintained for around 30 h, followed by the gradual degradation of $J_{ph}$ (Fig. 1d). According to the gas chromatography analysis for the *CG-400 μm* and *CG-1200 μm*, the cryogel-coated PEC devices produced the hydrogen gas at a Faradaic efficiency close to 100 % except the initial $J_{ph}$ drop period (Supplementary Fig. 6).

$J_{ph}/J_0$ of *CG-400 μm* reached 20% at 120 h and that of *CG-800 μm* and *CG-1200 μm* decreased to 30% at 210 h. This indicates that the thick overlayer significantly extended the lifetime of the PEC device compared to that of the reference device, and *no cryogel* showed full

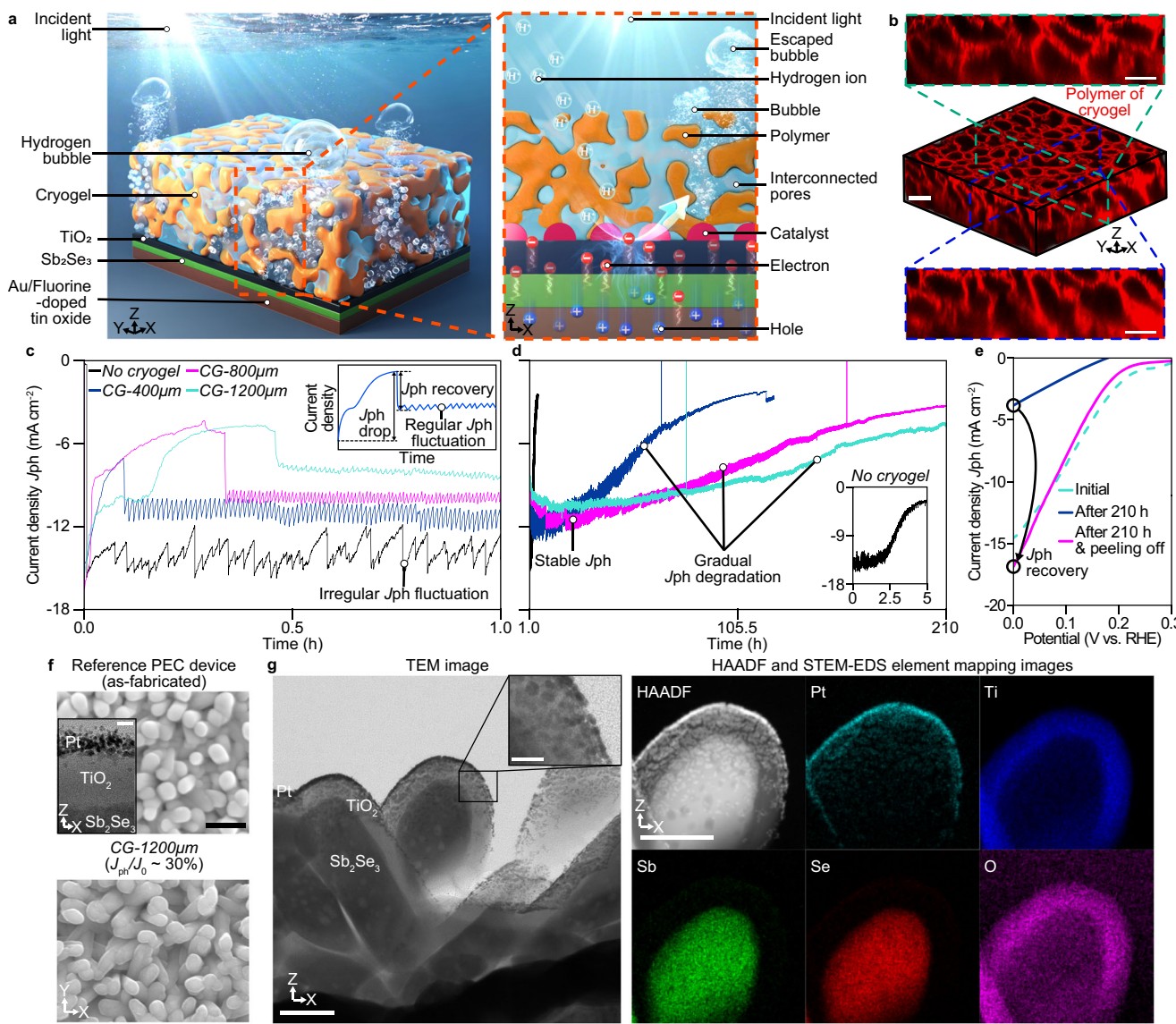

**Fig. 1 | Cryogelated polyacrylamide overlayer for photoelectrochemical (PEC) devices. a** Schematic illustration of the three-dimensional and cross-sectional structure of the device with a cryogelated overlayer. The schematic was drawn not to scale. **b** Three-dimensional fluorescent images and cross-sectional fluorescent images of the polymer structure of the cryogelated overlayer formed on the PEC device. The scale bar represents 50 μm. The current density ($J_{ph}$)-time profile of $Sb_2Se_3$ photocathodes with and without the cryogelated overlayer measured via chronoamperometry at 0 $V_{RHE}$ under 1-sun illumination during (**c**) 0–1 and (**d**) 1–210 h. The device without the cryogel was denoted as *no cryogel*. The device with the cryogel having thickness of 400, 800, and 1200 μm were referred to as *CG-400 μm*, *CG-800 μm*, and *CG-1200 μm*. The black, blue, magenta, and turquoise solid lines in (**c, d**) represent the $J_{ph}$-time profile of *no cryogel*, *CG-400 μm*, *CG-800 μm*, and *CG-1200 μm*, respectively. The inset graph in (**c**) indicates the schematic of the representative pattern of the $J_{ph}$-time profile of the PEC device with the cryogel overlayer during 0–1 h. The inset graph in (**d**) represents the $J_{ph}$-time profile of *no cryogel* during 0–5 h. **e** The $J_{ph}$-potential curve of *CG-1200 μm* measured via linear sweep voltammetry at the initial state (turquoise dotted line), after 210 h of stability testing (blue solid line), and after the mechanically peeling-off of the cryogelated overlayer (magenta solid line). **f** The surface microstructure of an as-fabricated reference PEC device and *CG-1200 μm* at a $J_{ph}/J_0$ of 30%, where $J_0$ represents the initial photocurrent density. The scale bar represents 500 nm. The inset image in the top image is the cross-sectional transmission electron microscopy (TEM) image of the reference PEC device. The scale bar in the inset image represents 10 nm. **g** Cross-sectional TEM image, high-angle annular dark field (HAADF), and scanning transmission electron microscopy-energy-dispersive X-ray spectroscopy (STEM-EDS) elemental mapping of the cross-section of *CG-1200 μm* at a $J_{ph}/J_0$ of 30%. The scale bars represent 100 nm. The scale bar in the inset of TEM image represents 20 nm.

$J_{ph}$ degradation within 5 h (inset of Fig. 1d) which was similar to the operation durations for other low-cost thin-film photocathodes (Supplementary Table 1). Compared to *CG-800 μm*, the current degradation occurred over the entire duration was reduced in *CG-1200 μm*. However, both devices exhibited a similar total gas production, which was higher than that of *CG-400 μm* (Supplementary Fig. 7). On the other hand, the initial $J_{ph}$ drop was not observed for *CG-100 μm* and *CG-200 μm*, which exhibited relatively poor device lifetimes (Supplementary Fig. 8a). *CG-100 μm* and *CG-200 μm* exhibited significant $J_{ph}$

degradation after 20 h of operation. Because the initial $J_{ph}$ drop was not observed and $J_{ph}$ exhibited an irregular fluctuation, the time-dependent $J_{ph}$ for thin overlayers was similar to that of the device without an overlayer. We observed local delamination of the 200 μm-thick overlayer after around 5 minutes of operation (Supplementary Fig. 8b). This result indicates that such a thin overlayer cannot withstand the mechanical stress developed during bubble growth[25], and its protective role might be diminished by physical delamination occurring at the early stage of operation. In our previous study, we reported

the 100 h stability ($J_{ph}/J_o \sim 70\%$) of a Pt/TiO$_2$/Sb$_2$Se$_3$ photocathode using a nanoporous polyacrylamide hydrogel protector that does not possess interconnected micropores for bubble escape[25]. An increase in the thickness of the hydrogel protector induced severe bubble accumulation, which increased the $J_{ph}$ degradation rate during long-term operation. Despite the microporous cryogelated overlayer helped gas transport, bubbles remained trapped in the overlayer could lead to reduction of current density.

Surprisingly, after mechanically peeling off the overlayer from *CG-1200 μm* after 210 h operation, $J_{ph}$ recovered from -5 mA cm$^{-2}$ to 16 mA cm$^{-2}$ for 0 V$_{RHE}$, as characterized by linear sweep voltammetry (Fig. 1e). Similarly, *CG-400 μm* and *CG-800μm* showed significant recovery of $J_{ph}$ after mechanically peeling off the overlayer (Supplementary Fig. 9a and b). In contrast, the onset potential and photocurrent of the *no cryogel* sample were significantly diminished after the stability test for 5 h (Supplementary Fig. 9c). This might be attributed to the detachment of the Pt catalyst from the device with the chemical dissolution of TiO$_2$/Sb$_2$Se$_3$ nanorods[10,25] (Supplementary Fig. 9d, e). We observed that the surface microstructures of *CG-400 μm*, *CG-800 μm*, and *CG-1200 μm* were maintained intact, although $J_{ph}/J_o$ was reduced to 30% (Fig. 1f and Supplementary Fig. 9f, g). Scanning transmission electron microscopy (STEM) and energy-dispersive X-ray spectroscopy (EDS) analyses of the cross-sectioned Sb$_2$Se$_3$ photocathode after peeling off the 1200 μm-thick overlayer showed that uniform Pt nanoparticles were clearly observed on the TiO$_2$/Sb$_2$Se$_3$ nanorods even after photocurrent degradation (Fig. 1g). This result suggested that the direct spatial confinement of catalyst nanoparticles by the nanoporous overlayer[25] might not play a critical role in stabilizing the catalysts. The TiO$_2$ layer of the *CG-1200 μm* experienced an approximate 8% decrease in thickness (Supplementary Fig. 10) due to the reductive dissolution. The crystallinity and orientation of Sb$_2$Se$_3$ were maintained, and no secondary phases were observed even after the PEC operation over 200 h (Supplementary Fig. 11). These observations raise two important questions: what is the origin of the gradual $J_{ph}$ degradation and what is the mechanism over which the cryogelated overlayer protects the surface microstructure of the device?

**Mechanical interaction between bubble and cryogelated overlayer**

In the experiments for devices with overlayers of various thicknesses, we determined that the 400, 800, and 1200 μm-thick cryogelated overlayers could reduce the surface structural damage of the device compared to devices without an overlayer but the photocurrent of the device was still degraded. We found that both the fluctuation amplitude and degradation rate decreased with the overlayer thickness (Supplementary Fig. 12). Since the regular $J_{ph}$ fluctuation was observed only after the drop and recovery of $J_{ph}$, the bubble dynamics including expansion and escape occurred during the $J_{ph}$ drop and recovery, respectively, can be critical in determining the photocurrent of the device with a cryogelated overlayer.

To determine the bubble dynamics, including nucleation, expansion, and escape during the initial $J_{ph}$ drop and the following recovery and fluctuation of $J_{ph}$, we visualized bubbles in the cryogelated overlayer in situ for *CG-800 μm*, which showed a large initial $J_{ph}$ drop (Fig. 2a). At the early stages of the PEC operation, small gas bubbles with different shapes and positions were observed due to the heterogeneity in shape and dimension of pores (Fig. 1b). Those bubbles were then merged into large bubbles at the later stage of operation. The large bubble might lead to the reduction of $J_{ph}$, as observed within 0.5 h after operation, via hindering the electrolyte transport and light transmittance (Fig. 1c). The electrochemical surface area measured before and after the bubble formation was similar (Supplementary Fig. 13), suggesting that the bubble was nucleated at a site away from the device surface. The large trapped bubbles were further expanded by the continuous gas production, deforming the overlayer. This

suggests that the hydrogen ions could be still transported through the porous structure of the cryogel even in the presence of large bubbles (Supplementary Fig. 14). When the bubble size reached a critical value, a bulge, which is a local deformation with a high curvature, of the overlayer was observed. Formation of the bulge and the following escape of bubbles through it occurred less than 0.1 s indicative of local fracture in the cryogel structure (Supplementary Fig. 15). The size of the trapped bubble in the overlayer was reduced by the bubble escaping instantaneously, resulting in the initial $J_{ph}$ recovery. The size of the escaped bubbles was smaller in the repeated cycles than in the initial escape (Fig. 2a), which was consistent with the reduction in the amplitude of the $J_{ph}$ fluctuation relative to the initial drop and the following recovery of $J_{ph}$.

To determine if the relatively small fluctuation of $J_{ph}$ in the repeated cycles could be attributed to the local fracture in the overlayer, we investigated the mechanical stress produced during the expansion of trapped bubble using finite-element method (FEM). In the FEM model, the initial size of a bubble trapped in an overlayer was based on the dimensions observed in the experiments. The transport of gas molecules from the PEC device to the bubble was simulated by air injection through the inlet at the backside of the model, which was a fixed boundary. The bubble expanded to the front side of the overlayer, which was a free boundary (Fig. 2b). We determined that the central region of the bubble expanded more than the edges, as observed in the experiments. A map of the von Mises stress revealed that the stresses were localized at the bubble edges where bulging was observed in the experiment. We further analyzed the stress distribution in the overlayer with various overlayer thicknesses. The bubble area and overlayer thickness were normalized by the area and thickness of the initial bubble, respectively. The stress concentrated at the edge of the bubble increases as the overlayer thickens (Fig. 2c). The increase in the maximum stress proportional to the overlayer thickness was more significant at higher bubble expansion ratios (Fig. 2d). For the relative bubble size equal to 4, the maximum stress slightly decreased as the normalized thickness exceeded 5. The high mechanical stress produced, especially at the bubble edge, could increase the probability of fractures enlarging the local porous area, which is more significant at a thick overlayer (Fig. 2e)[33,34]. Initiation and propagation of fracture in the overlayer is stochastic, and it depends on the overlayer thickness, resulting in the difference in the timepoints of the photocurrent recovery for different thicknesses (Fig. 1c).

To investigate the effect of the overlayer fracture on the current fluctuation, we theoretically analyzed how the bubble dynamics are altered by changes in the pore structure through operation (Fig. 3a and Supplementary Note 1). The volume ($V_{bubble}$) of the gas bubble was increased by the constant production of hydrogen gas. The bubble expansion increases the internal bubble pressure ($P_{bubble}$) and pore size ($\xi_{pore}$). An increase in $\xi_{pore}$ decreases the capillary pressure ($P_{cap}$) of the surface pore, according to the Laplace–Young equation. When $P_{bubble}$ exceeded $P_{cap}$, the bubbles escaped from the overlayer surface. Bubble escape can occur in two scenarios, namely with and without the mechanical fracture of the overlayer. $\xi_{pore}$ in the overlayer is assumed to vary proportionally with the size of the bubble owing to the global deformation of the overlayer (Fig. 3b, top). When $P_{bubble}$ reaches a critical pressure $P_{fracture}$, irreversible mechanical damage of the fracture occurs at the bulge in the overlayer, which leads to a local break and the consequential coalescence of pores (Fig. 3b, bottom). In scenario 1, where no fracture occurs, part of the trapped bubble escapes when $P_{cap}$ is decreased to be equal to $P_{bubble}$. Owing to the bubble escape, $V_{bubble}$ decreases, resulting in a reduction in $P_{bubble}$, while $P_{cap}$ increases (Fig. 3c, left). The sawtooth-like cycles of $P_{bubble}$ and $P_{cap}$ were repeated via continuous gas production. In scenario 2, where the fracture occurs at $P_{fracture}$, $\xi_{pore}$ can increase owing to the local fracture, leading to a sudden reduction in $P_{cap}$ (Fig. 3c, right). The bubbles continuously escape until $P_{bubble}$ becomes smaller than

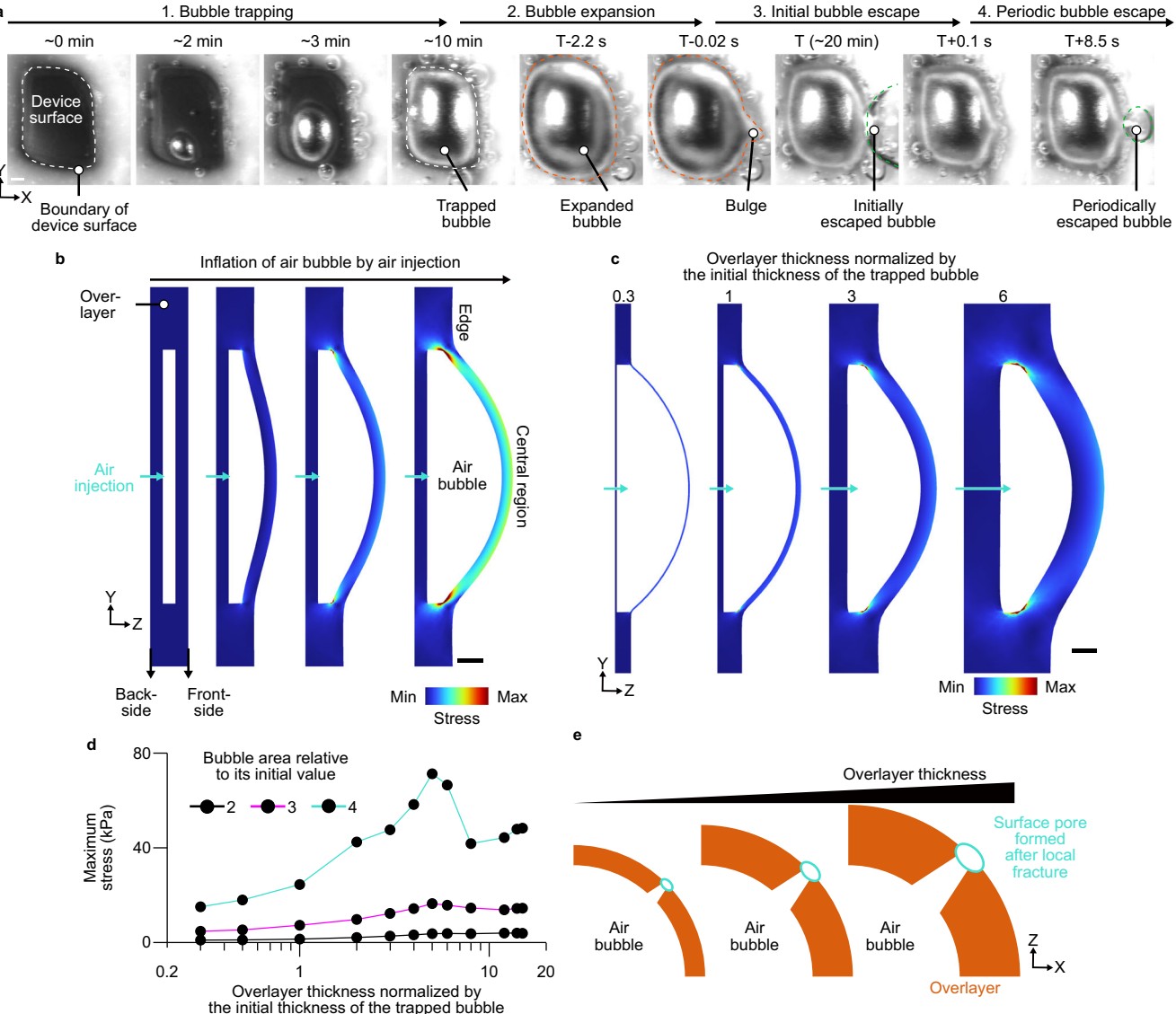

**Fig. 2 | Effect of cryogelated overlayer on the durability of Sb₂Se₃ photo-cathodes. a** Time-lapse images of trapping, expansion, and escape of bubbles in the device with 800 μm-thick cryogel *CG-800 μm*. White dotted lines in the images represent the boundary of the device surface. The orange and green dotted lines indicate the boundary of the expanded bubbles before the initial bubble escape and the escaped bubbles. The time at the bubble escape was denoted as 'T'. **b** Finite-element method (FEM) result of the stress on the overlayer by the inflation of the air bubble trapped inside the overlayer. **c** FEM result of the stress applied on the overlayer with varying thicknesses by the expanding bubble. The overlayer thickness normalized by the initial thickness of the air bubble in the overlayer was 0.3, 1, 3, and 6. The turquoise arrows in (**b**, **c**) represent the injection of the air into the trapped bubble. The color scale in (**b**, **c**) represents the normalized stress applied on the overlayer. **d** Maximum stress at the bubble-overlayer interface at various overlayer thicknesses normalized by the initial thickness of the trapped bubble when the relative area of the bubble to its initial value was 2 (black), 3 (magenta), and 4 (turquoise). **e** Schematic of the surface pore formed by the local fracture in the overlayer with varying thicknesses. The white solid line in the first image of (**a**) and black solid lines in (**b**, **c**) are the scale bars. The scale bar in (**a**) represents 0.5 mm and those in (**b**, **c**) represent 1 mm.

$P_{cap}$. Owing to the plastic increase of $\xi_{pore}$ by fracturing, $P_{cap}$ after the initial bubble escape cannot recover its original value and instead fluctuates with an average lower than that without the fracture. As a result, the following bubble escape occurred at a lower $P_{bubble}$ compared to that without fracture.

### Effect of bubble dynamics on structural stability of photoelectrode

To investigate the effect of the fracture degree on bubble fluctuation in the overlayer, we analyzed $P_{bubble}$ and $V_{bubble}$ as a function of the degree of pore coalescence $\lambda_{pore}$ (Fig. 3d), defined as the ratio of the pore size after fracturing $\xi_{pore,2}$ to the original pore dimension $\xi_{pore,1}$, as shown in Fig. 3b. At $\lambda_{pore} = 1$, $P_{bubble}$ and $V_{bubble}$ begin to fluctuate once they reach each critical value as no fracturing occurs. In contrast,

when $\lambda_{pore}$ is greater than 1, $P_{bubble}$ and $V_{bubble}$ exhibit a sudden reduction prior to fluctuation owing to an instantaneous enlargement of $\xi_{pore}$ (Supplementary Fig. 16). The time-dependent fluctuating patterns of $P_{bubble}$ and $V_{bubble}$ were similar to the cyclic variation in the photocurrent observed in the experiments (Fig. 1c). This result indicates that the large initial $J_{ph}$ recovery followed by a relatively small $J_{ph}$ fluctuation can be attributed to the local fracture caused by the stress localized in the overlayer.

For escaping of trapped bubbles, they should be transported through not only fractured pore but micropores present in the path for gas transport. We could observe the gas transport through the layer of micropores present between the trapped bubble and the surface of the cryogel overlayer (Supplementary Fig. 17a). At a given pressure difference in a channel, the gas transport rate is inversely proportional to

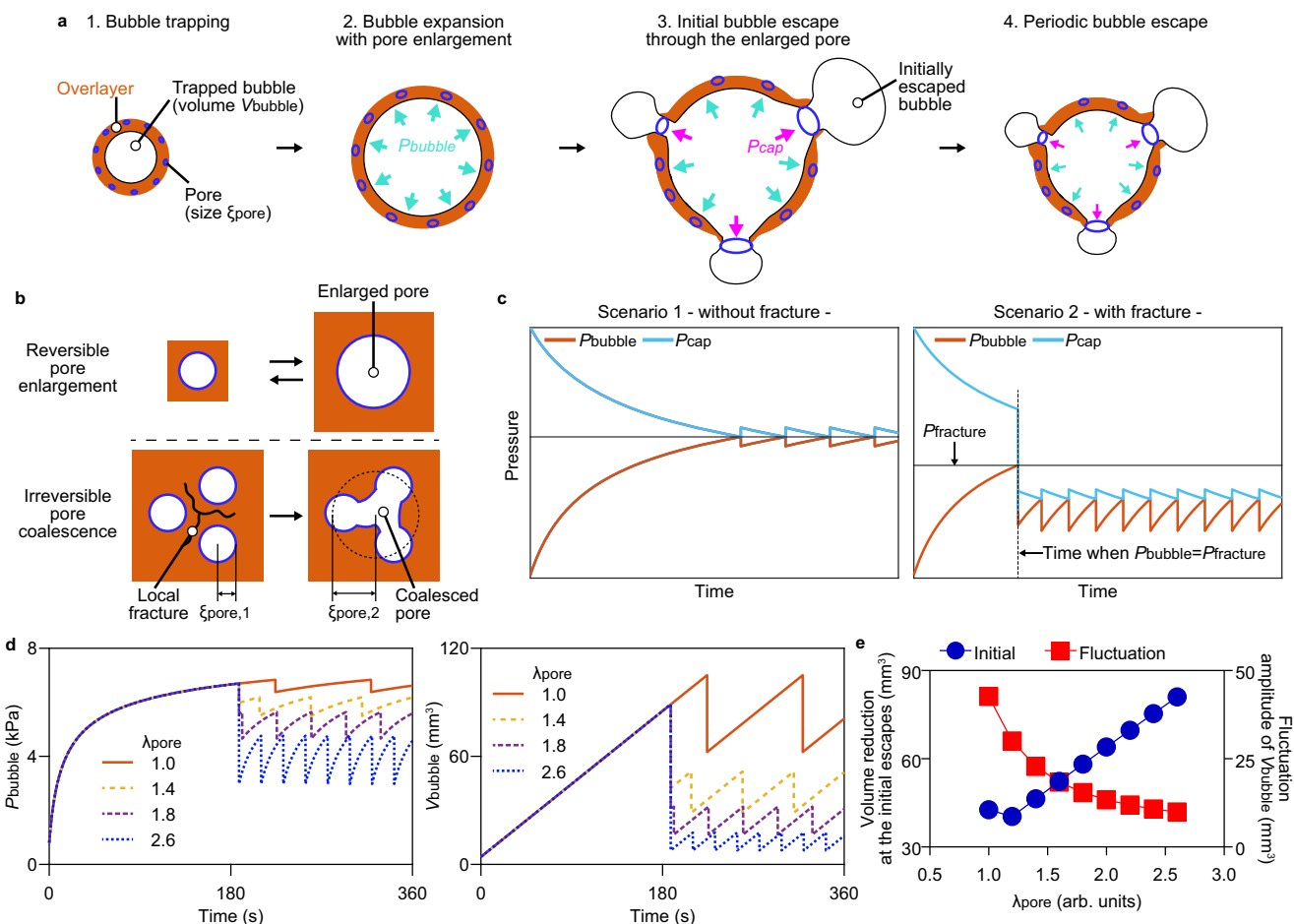

**Fig. 3 | Bubble escape dynamics in the cryogelated overlayer. a** Schematic of bubble expansion in and escape from the overlayer. **b** Schematical illustration of the reversible and irreversible pore enlargement. The pore radius after the fracture $\xi_{pore,2}$ is defined as the effective radius of coalesced pores. **c** Schematical representation of the time-dependent internal bubble pressure $P_{bubble}$ (orange solid line) and capillary pressure $P_{cap}$ (light-blue solid line) without and with the local fracture.

**d** Theoretical analysis of $P_{bubble}$ and gas bubble volume $V_{bubble}$ at various degrees of pore coalescence $\lambda_{pore}$. The orange solid lines, yellow dotted lines, purple dotted lines, and blue dotted lines represent that $\lambda_{pore}$ is 1.0, 1.4, 1.8, and 2.6, respectively. **e** Volume reduction at the initial escape (blue circle) and fluctuation amplitude (red rectangle) of $V_{bubble}$ as a function of $\lambda_{pore}$.

the path length ($L_{path}$), that can increase in the thicker overlayer, according to Poiseuille's Law[35]. Additionally, the resistance for gas transport can be increased by the elastic expansion and contraction of the pores. The distensibility of cryogel should be taken into account to estimate an effective value of $L_{path}$. The reduction in the $V_{bubble}$ at the initial escape increased with $\lambda_{pore}$ (Fig. 3e) but decreased with the effective $L_{path}$ (Supplementary Fig. 17b). The fluctuation amplitude of the $V_{bubble}$ was decreased by both $\lambda_{pore}$ and effective $L_{path}$ (Fig. 3e and Supplementary Fig. 17b). This result is consistent with the experimental result that the photocurrent fluctuation decreased with increasing overlayer thickness (Supplementary Fig. 12).

Long-term monitoring of the bubble for *CG-400 μm*, where the fluctuation amplitude and degradation rate of $J_{ph}$ were large, revealed that the size of bubble decreased with cycles of $J_{ph}$ fluctuation (Fig. 4a). This might be attributed to the fatigue-induced softening of the overlayer, which could occur because of the cyclic deformation by the bubble. Theoretical analysis showed that the size of the trapped bubble can be decreased when the elastic modulus of the overlayer is reduced to a critical value by softening that occurs through cycles (Supplementary Fig. 18). Images of the surface structure after long-term operation over 150 h showed that the surface structure was more intact in the center of the bubble than its edge (Fig. 4b). The structural degradation might be attributed to the shear strain produced,

especially at the edges, owing to the cyclic contraction and expansion of the bubble (Supplementary Fig. 19a, b). The shear stress on the catalyst on the device surface could lead to detachment and agglomeration of the Pt catalyst[16] and reductive dissolution of the $TiO_2$ layer[29]. As fatigue-induced softening depends on the strain amplitude[36], it would probably occur more for a thin overlayer where a relatively large $J_{ph}$ fluctuation is observed. Compared to *CG-400 μm*, a decrease in the bubble size with cycling was not observed for *CG-1200 μm* during long-term operation (Supplementary Fig. 19c). We observed a large number of microbubbles at the boundary of the large bubble in the overlayer for *CG-1200 μm* during the long-term operation (Fig. 4c). The large trapped bubbles in the thick cryogel not only migrate toward surface pores for escape but also enter adjacent micropores. Those gas bubbles were unable to easily pass through the pores and could be trapped to form microbubble clusters due to their aerophobicity. In contrast to large bubbles, microbubbles in the cryogel did not seem to damage the structure of Pt catalyst on the device surface (Fig. 1g). We speculated that the mechanical interaction with the cryogel occurred during bubble dynamics and consequential shear stress on the device surface would be less for microbubbles confined at a distance from the device surface due to their smaller dimension. The accumulation of trapped microbubbles in the overlayer by its microporous structure (Fig. 4d) can hinder not only the mass transfer of electrolytes but also light

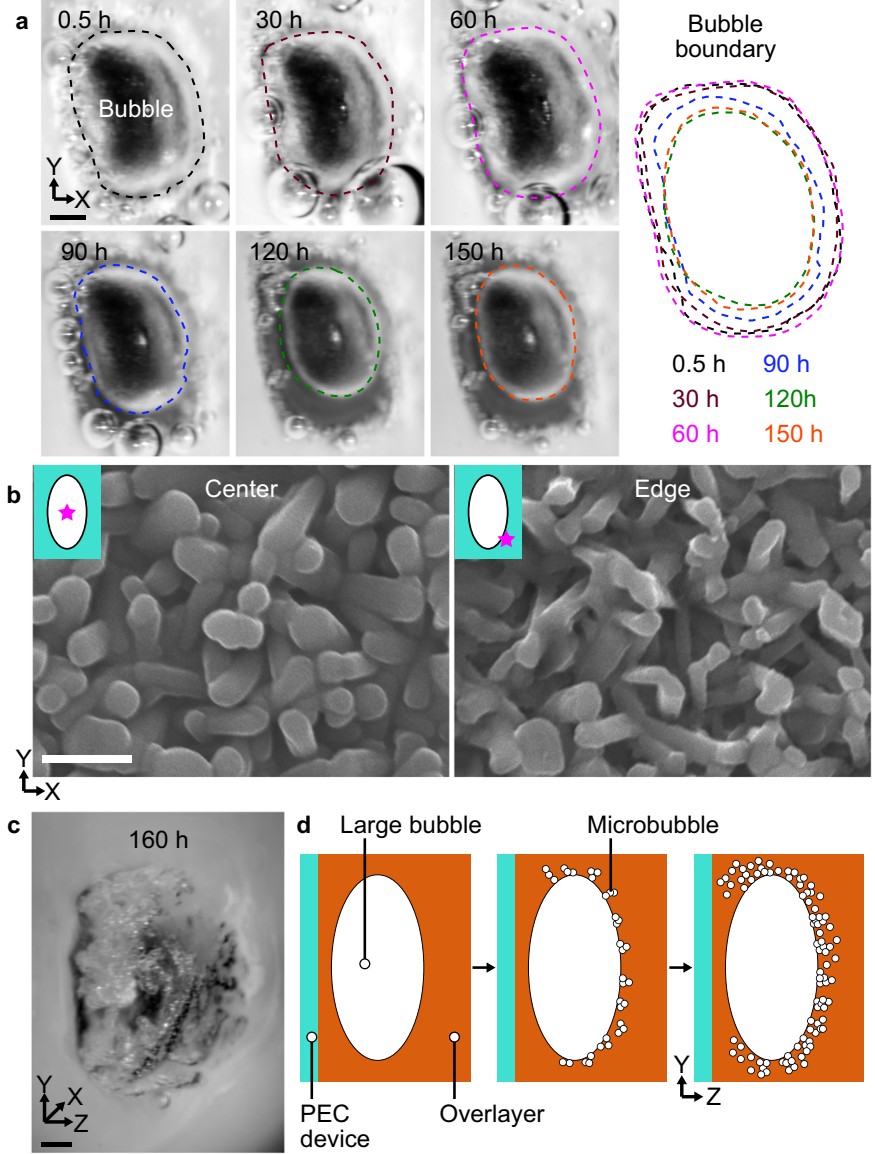

**Fig. 4 | Distinct mechanism of the photocurrent degradation in *CG-400 μm* and *CG-1200 μm*. a** Photographs of the bubble trapped in the overlayer of *CG-400 μm* after the partial bubble escape at 0.5, 30, 60, 90, 120, and 150 h. The black, brown, magenta, blue, green, and orange colored dotted lines represent the boundary of the trapped bubbles at 0.5, 30, 60, 90, 120, and 150 h, respectively. **b** Scanning electron microscopy images of the surface of *CG-400 μm* in the center and edge of the bubble after operation of 150 h. In the inset schematics for SEM images, the white ellipse, turquoise square, and magenta star represent the trapped bubble, PEC device, and location at which each image was captured, respectively. **c** Photograph of the bubbles at 160 h in *CG-1200 μm*. **d** Cross-sectional schematic of the trapping of microbubbles in the region around the bubble during the long-term photoelectrochemical (PEC) operation. The scale bars in (**a**, **c**) represent 1 mm and that in (**b**) represents 500 nm.

transmission onto the device surface, leading to a gradual reduction in the photocurrent of the device. Our results showed that the thicker cryogel overlayer could increase the structural stability while reducing the current density of the device. Thus, to obtain the optimal thickness of cryogel requires consideration in terms of lifetime and photo-current density.

## Discussion

We revealed that the microporous structure of cryogelated overlayer plays an important role in enhancing the stability of PEC device by regulating dynamics of produced hydrogen bubbles. To further investigate the effect of the pore structure of cryogel on the device stability, we prepared PEC devices incorporated with cryogelated overlayers having disconnected micropores and interconnected micropores incorporated with macropores at the top surface of the overlayer which are denoted as *disconnected micropores* and *surface macropores*, respectively (see details in Supplementary Note 2). For *disconnected micropores*, the initial $J_{ph}$ drop to 5 mA cm$^{-2}$ was observed without recovery and a low $J_{ph}$ was maintained for 30 h (Fig. 5a). Initially, small bubbles accumulated on the surface of the device after only ~30 s of operation, then grew along the device surface and merged into a large bubble after ~7 min. This large bubble did not show a noticeable change in size during the continuous PEC operation, indicating that partial escape of the trapped bubble occurred. How-ever, after the operation, some bright spots were observed in the scanning electron microscopy (SEM) images of the device surface. This result indicated that the Pt particles were continuously aggregated to form large particles while the Pt detachment was suppressed by the overlayer. It is speculated that the expansion of bubbles nucleated in the isolated micropores near the device surface can generate

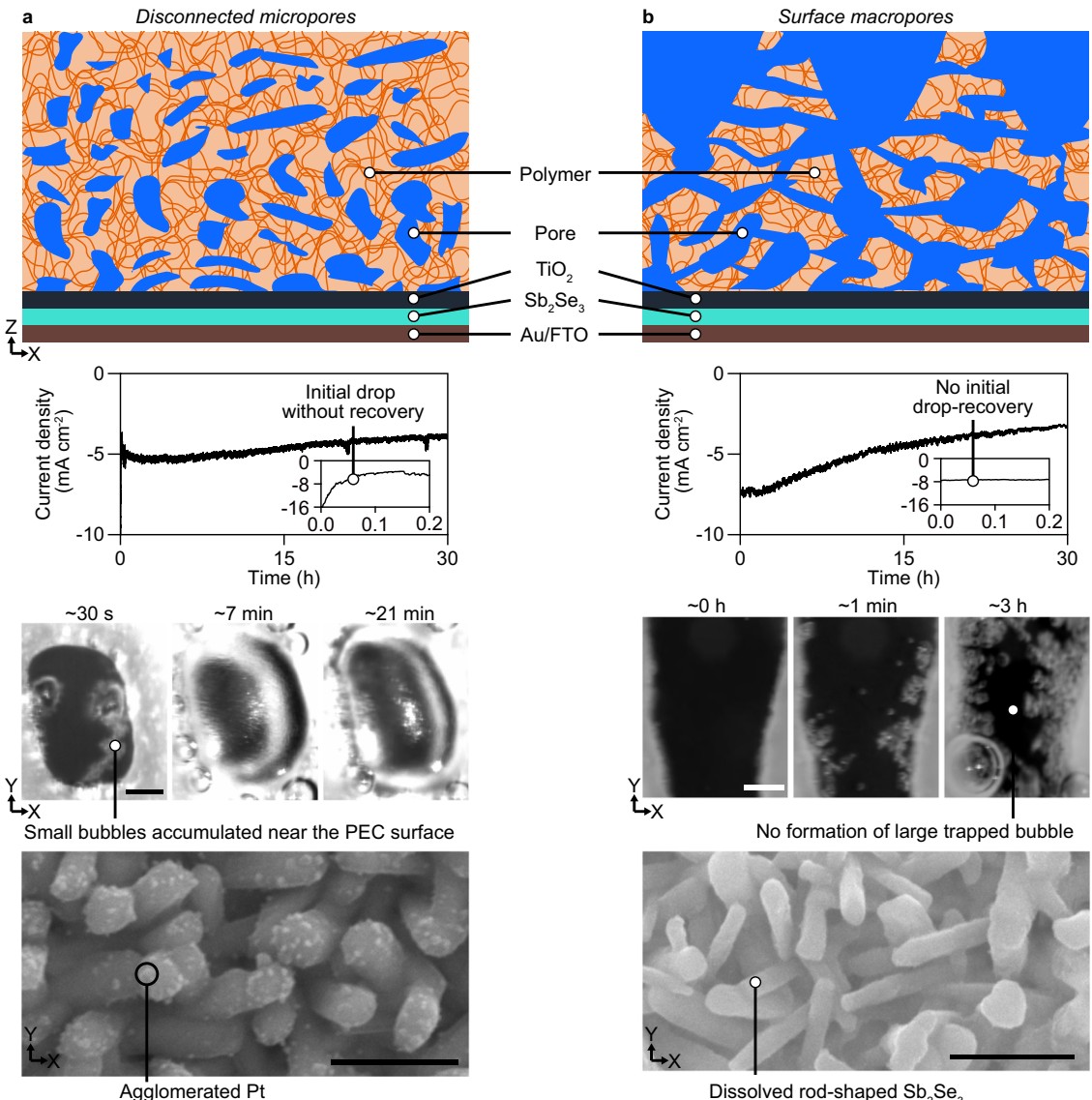

**Fig. 5 | Effect of the microstructure of the cryogelated overlayer on the functional and structural stability of the photoelectrochemical (PEC) device.** Schematic of the cross-sectional structure of the as-fabricated device with an overlayer, photocurrent density of the protected PEC device, photographs of bubbles in the protector, and scanning electron microscopy (SEM) images of the device surface of (**a**) *disconnected micropores* and (**b**) *surface macropores*. The inset graphs represent the photocurrent density from 0 to 0.2 h. The scale bars in the SEM images represent 500 nm and those in the photographs of the bubbles represent 1 mm.

mechanical stress to the PEC device (Supplementary Fig. 20). Bubbles accumulated on the device surface may also hinder photocurrent generation by blocking the active sites of the catalyst. In contrast, when micropores are interconnected inside the overlayer, hydrogen gas might translate first, followed by bubble nucleation at the top part of the overlayer, thereby maintaining the structural stability of the device. For *surface macropores*, $J_{ph}$ gradually decreased to ~50% of its initial value after 30 h (Fig. 5b). With intrinsic macropores at the top surface of the overlayer, the microbubbles effectively escaped rather than formed a large bubble in the overlayer. We observed significant dissolution of the rod-shaped $Sb_2Se_3$, similar to the device without a cryogelated overlayer, indicating that structural degradation occurred without initial bubble formation. This might be due to the detachment and agglomeration of the Pt catalyst[10] and the reductive dissolution of the $TiO_2$ layer. This result indicates that the structural protection of the PEC device by the cryogel requires the nucleation and trapping of bubbles at the region away from the device surface. The bubble trapped in the overlayer with interconnected micropores might act as an

artificial bubble nucleation site, which can absorb gas molecules[37]. Suppression of bubble nucleation at the device surface might contribute to preserving the Pt co-catalyst surface. The bubble dynamics in the cryogel overlayer including nucleation, coalescence, trapping, and escape of bubbles, which are critical in determining the protective function of the cryogel, can be regulated by the wettability of the porous material[38].

We expect that our cryogel overlayer protection would also be beneficial for the photoanode side when the co-catalyst is chemically stable in the electrolyte. The pH of the electrolyte may affect the swelling of the hydrogel, leading to changes in the structural properties of the gel, such as pore size and thickness. By incorporating the internal porous structure and dimension tailored specifically to the operation conditions of the device, the cryogel overlayer would have the potential to enhance the structural stability of the PEC devices, including photoanodes decorated with chemically stable co-catalysts.

Since the $J_{ph}$ of *CG-1200 μm* was reduced without structural damage of the device, we hypothesized that the $J_{ph}$ can be recovered

when the trapped bubbles are effectively removed. It is assumed that bubbles can be removed by the spontaneous diffusion of gas molecules when we periodically stop the PEC operation for the cessation of gas production. We performed a chronoamperometry test with light chopping, repeating 3 h of light-on and 1 h of light-off cycle (Supplementary Fig. 21). For the first few cycles, the $J_{ph}$ was almost recovered after each light-off cycle albeit it gradually decreased because the bubbles were not completely removed. This result clearly demonstrates that the PEC performance can be recovered by removing trapped bubbles. To test the device in a more practical condition, we simulated the daily day and night cycle (see details in Supplementary Note 3). We observed that microbubbles trapped in the overlayer during the day cycle can escape during the dark cycle. Despite the bubble removal, $J_{ph}$ degradation did not recover. It might be due to the irreversible interface oxidation as evidenced by the sudden change from the negative to positive dark potential during the night cycle.

We expect that our cryogelated overlayer can be widely applied to various TiO$_2$-protected PEC devices because bubble growth along the device surface is well prevented and bubble transport through the interconnected micropores of the overlayer is independent of the type of light-absorbing material. Furthermore, we aimed to achieve the large-scale fabrication of photoelectrodes to realize the PEC system. Since previously reported catalyst protection strategies typically deposit thin materials (<10 nm), it would be very challenging to maintain the uniformity of such a thin layer on large-scale devices. In contrast, our cryogelated overlayer technique does not suffer from thickness uniformity, which is a great advantage for large-scale applications. In particular, applying this strategy to solution-processed thin-film-type photoelectrodes would be more suitable for practical applications in economical carbon-free hydrogen production.

In this study, we investigated the protection mechanism for overlayers with porous structure by utilizing engineered cryogel overlayers. We found that bubbles trapped near the top of the cryogel during the early PEC operation function as regulated bubble nucleation sites, providing a temporal pathway for transporting gas molecules to the outer electrolyte solution. As a result, the mechanical shear stress produced by bubbles at the device surface can be reduced. Employing the 1200 μm-thick cryogelated overlayer, the structure of the Pt/TiO$_2$/Sb$_2$Se$_3$ photocathode remained intact after 210 h operation, exhibiting approximately 30% $J_{ph}/J_o$. Through the experimental, numerical, and theoretical analysis about the mechanical interaction between the bubble and the cryogelated overlayer, we found that the reduction of cyclic expansion of the overlayer by bubbles was critical in enhancing the device lifetime. Our study will be beneficial for designing a semi-permanent green hydrogen-producing device that is essential for a carbon-neutral society.

## Methods

### Materials
1 N sodium hydroxide (NaOH, 98%; UN1823), ethyl alcohol (99.9%; UN1170), and acrylamide (AAM, 98%; A503181), were purchased from Duksan (Ansan, Korea). Glutaraldehyde (50%; 4133-1405) was purchased from DaeJung (Siheung, Korea). N,N'-methylenebisacrylamide (BAAM, 99%; 146072), ammonium persulfate (AP, ≥98%; 248614), N,N,N',N'-tetramethylethylenediamine (TEMED, 99%; T9281), and (3-aminopropyl)triethoxysilane (APTES, ≥98%; A3648) were purchased from Sigma-Aldrich (St. Louis, MO, USA).

### Fabrication of Pt/TiO$_2$/Sb$_2$Se$_3$/Au/FTO devices
A 70 nm-thick Au layer was deposited on the back-contact substrate via thermal evaporation on fluorine-doped tin oxide (FTO) glass (TEC-15, Pilkington, UK). The Sb$_2$Se$_3$ absorbing layer was fabricated by a simple spin-coating and annealing process conducted inside an N$_2$-filled glovebox. We prepared Sb ink by dissolving 0.258 g of SbCl$_3$ in 12 ml of 2-methoxyethanol (2ME). Se ink was prepared by dissolving 0.385 g of

Se in a solvent mixture of 0.45 ml of thioglycolic acid (TGA) and 7.37 ml of ethanolamine (EA). The two prepared inks were homogeneously mixed at 80 °C overnight prior to spin-coating. The mixed solution was spin-coated at 2000 rpm for 30 s, followed by drying at 180 and 300 °C for 3 min each. After repeating this spin-coating and drying process ten times, additional annealing was conducted at 350 °C for 20 min. Subsequently, the sample was post-annealed at 200 °C for 30 min in air, followed by direct deposition of a TiO$_2$ protection layer through ALD (NCS Inc., Daejeon, Korea). A total of 600 ALD cycles were performed at 120 °C using tetrakis(dimethylamido)titanium (TDMAT) and H$_2$O as the Ti and O precursors, respectively. Each cycle comprised a TDMAT pulse (0.3 s) followed by 15 s of N$_2$ purging, and an H$_2$O pulse (0.2 s) followed by 15 s of N$_2$ purging. The growth rate of TiO$_2$ was approximately 0.58 Å/cycle. Finally, the Pt co-catalyst was sputtered using a sputter coater (Ted Pella, Redding, CA, USA) at an applied current of 10 mA for 120 s. The Pt/FTO electrodes were also fabricated using the same deposition procedure to characterize the optical, structural, and electrochemical properties of Pt co-catalyst layer prior to coating onto the photoelectrodes. (Supplementary Note 4).

### Deposition of the cryogelated overlayer on devices
The as-prepared Pt/TiO$_2$/Sb$_2$Se$_3$/Au/FTO device was treated with NaOH, APTES, and glutaraldehyde to form chemical bonds with the overlayer. The device was dipped in a 0.1 N NaOH solution for 5 min, treated with a 0.5% APTES solution for 5 min, and then washed with ethyl alcohol. Subsequently, the device was treated with a 0.5% glutaraldehyde solution for 30 min, washed with deionized (DI) water, and dried at 60 °C in an oven. The self-assembled monolayer (SAM) of APTES-glutaraldehyde is widely used to immobilize the polyacrylamide hydrogel or proteins on the TiO$_2$ or SiO$_2$ substrate[39–41]. The silanol group of hydrolyzed APTES reacts with the hydroxyl group of the substrate formed by the NaOH treatment. The amine group of APTES then reacts with the aldehyde group of glutaraldehyde. During the gelation of the polyacrylamide hydrogel, the polymer network of polyacrylamide can be combined with another end of glutaraldehyde. However, this monolayer did not significantly influence the PEC performance without the cryogel overlayer (Supplementary Fig. 22). Without this SAM treatment step, the cryogel was easily detached from the device.

For the fabrication of rubber molds that control the thickness of the cryogelated overlayer, a commercial rubber film (ELASTOSIL 2030, Wacker, Munich, Germany) was cut using a cutting plotter instrument CE6000-120 (Graphtec, Yokohama, Japan). The rubber molds were then placed on PEC devices treated with APTES and glutaraldehyde.

DI water, a 100% w/v (weight/volume percentage) AAM solution, and a 2% w/v BAAM solution were mixed by gentle pipetting followed by degassing for 1 h. The addition of 10% AP and TEMED solutions to the mixture initiated free-radical polymerization. The pre-gel solution was injected into the rubber mold on the PEC device and enclosed by a cover glass to prevent liquid evaporation during cryogelation and oxygen-induced inhibition of free-radical polymerization. 1 ml of pre-gel solution was prepared by mixing 888 μl of DI water, 65.2 μl of 100% AAM, 40.8 μl of 2% BAAM, 5 μl of 10% AP, and 1 μl of TEMED. The cryogelation process was conducted at the same location with a minimal temperature gradient in a freezer. After cryogelation in a −20 °C freezer in the laboratory for 12, 24, 48, or 72 h, the rubber mold and cover glass were carefully removed from the cryogelated overlayer. To facilitate detachment of the cover glass without causing any damage to the overlayer, the cover glass was treated with a water-repellant agent (Rain-X, Illinois Tool Works, Illinois, USA) for 3 min before initiating gelation. Finally, the surface of the PEC device without the overlayer was covered with an epoxy resin.

Because the as-fabricated polyacrylamide cryogel on the slide glass had low optical transmittance, which is not suitable for photoelectrochemical (PEC) applications, we added glycerol as an

anti-freezing agent to the pre-gel solution (Supplementary Fig. 3a). The optical transmittance was improved by increasing the glycerol concentration from 2.5% v/v (volume/volume percentage) to 5%v/v. However, further addition of glycerol to 10% v/v did not significantly increase the transmittance. Increasing the monomer concentration of the cryogel from 6% w/v to 30% w/v significantly reduced the light transmittance for all wavelengths (Supplementary Fig. 3b). When the monomer concentration of the pre-gel solution was lower than 5% w/v, the cryogel protector was easily damaged during the detachment of the cover after cryogelation, owing to its low mechanical strength. Therefore, we selected a glycerol concentration of 5% v/v and a monomer concentration of 6% w/v for the fabrication of the cryogel on the PEC device.

### Characterization of the cryogelated polyacrylamide overlayer
To characterize the porous structure, the overlayer samples were fabricated between two Rain-X-treated glass substrates and then cut into a cylindrical shape with a diameter of 4 mm using a biopsy punch. Each sample was immersed in a 2% w/v solution of carboxylate-modified polystyrene (PS) fluorescent NPs with a diameter of 200 nm (Invitrogen, Carlsbad, CA, USA) for 24 h. The carboxylate-modified NPs could be attached to the amide group of the polymer network of the polyacrylamide hydrogel and cryogelated sample via hydrogen bonding. The unbound PS NPs were removed from the pores of the cryogelated polyacrylamide layer by washing the samples with DI water (Duksan Pure Chemicals, Ansan, Kyungkido, Korea). Three-dimensional fluorescent images of the NPs in the samples were captured using a confocal fluorescence microscope (LSM 880, Carl Zeiss, Oberkochen, Baden-Württemberg, Germany). The fluorescent image of the 200 nm carboxylated fluorescent NPs represents the structure of the polymer wall of the cryogelated sample.

For quantitative analysis of the pore structure, fluorescent images of the PS NPs inside the cryogelated sample were binarized. In the binarized images, the pores and polymer walls of the cryogelated polyacrylamide sample were manually segmented using Fiji software (National Institutes of Health, Bethesda, MD, USA). Based on the segmented images, the effective diameter of the micropores in the cryogelated polyacrylamide layer was calculated.

### Photoelectrochemical characterizations
PEC characterizations, including linear sweep voltammetry and chronoamperometry, were conducted in a typical three-electrode system with Ag/AgCl/KCl (saturated) reference and Pt counter electrodes. The three electrodes were submerged in an Ar-purged 0.1 M $H_2SO_4$ electrolyte (pH = 1), and a commercial AM 1.5 G solar simulator and a Si reference cell (Newport Corporation, USA) were utilized for the simulated solar light and 1-sun calibration, respectively. For all measurements, the applied potential was changed to the reversible hydrogen electrode (RHE) scale using the following equation: $E_{RHE} = E_{Ag/AgCl} + 0.059 pH + 0.19$. The practical day/night cycle test was performed by chronoamperometry at 0 $V_{RHE}$ under illumination for 12 h and the device was neglected under open-circuit potential in the dark for another 12 h (repeated).

### Materials characterization
Optical transmittance spectra were recorded at room temperature using a UV-vis spectrophotometer (V-670, JASCO, Easton, MD). The surface morphology of the $Sb_2Se_3$ photocathodes was analyzed using field-emission SEM (JSM-7001F, JEOL Ltd., Tokyo, Japan). A focused ion beam technique was used to prepare electron-transparent foils from the selected $Sb_2Se_3$ photocathode and the microstructures were analyzed using high resolution transmission electron microscope (Talos F200X, FEI, Hillsboro, OR, USA) at an acceleration voltage of 200 kV.

### Imaging of the dynamic behaviors of bubbles in the overlayer
A high-speed camera C320 (Vision Research, New Jersey, USA) and a digital single-lens reflex camera D5300 (Nikon, Tokyo, Japan) were used for imaging the bubble dynamics in the overlayer for short and long durations, respectively[25]. A 105 mm macro lens (Nikon, Tokyo, Japan) and a 2X teleconverter (TELEPLUS MC7 AF 2x DGX, Kenko, Tokyo, Japan) were attached to the cameras. Bubbles on the PEC device were captured without an additional light source.

### Numerical analysis of the overlayer and bubble
We developed a time-dependent FEM model to analyze the dynamic interaction between the overlayer and inflating bubble using the commercial FEM software COMSOL Multiphysics (COMSOL, Stockholm, Sweden). We made a two-dimensional cross-sectional model consisting of an overlayer and an air bubble trapped inside the overlayer. The dimensions of each domain were determined by considering those of the overlayer used in the experiment. Based on our experimental observations, the surface-covering bubble was placed in the middle region of the overlayer along the thickness direction.

The cryogelated overlayer was modeled as an isotropic linear elastic material. The elastic modulus of the overlayer was set at a constant value of 10 kPa. The backside of the overlayer, which was the boundary of the overlayer where the flow inlet was placed, was set as the fixed boundary. The front side of the overlayer, which is the boundary opposite the backside, was set as a free boundary.

The air gas bubble in the overlayer was modeled as an incompressible laminar flow. To model the inflation of the air gas bubble, we injected fluid through the inlet without an outlet into the air gas domain. The inlet of the air domain is placed outside the overlayer. The normal flow at the inlet was set as a fully developed flow at a constant speed. The tangential flow speed at the inlet was zero. The deformation of the cryogelated overlayer by the inflating gas was performed by applying the boundary condition of a fully coupled fluid–structure interaction at the interface between the solid and fluid domains and a deforming domain in the domain of the air bubble.

## Data availability
The data that support the findings of this study are available from the corresponding authors upon request. The data generated in this study are provided in the main article, Supplementary Information and Source Data file. Source data used to generate the graphs in Figs. 1c–e, 2d, 3d, e, 5a, b are provided in the Source Data file. Source data are provided with this paper.

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

## Acknowledgements

This work was supported by the National Research Foundation (NRF) of Korea (Nos. 2021R1A3B1068920 (J.M.), 2021M3H4A1A03049662 (J.M.), 2021R1A2C2009070 (Hyungsuk Lee), and RS-2023-00253895 (J.T.)) funded by the Ministry of Science and ICT. This research was also supported by the Yonsei Signature Research Cluster Program of 2021 (2021-22-0002 (J.M.)) and by the Technology Innovation Program (20013621, Center for Super Critical Material Industrial Technology (Hyungsuk Lee)) funded By the Ministry of Trade, Industry & Energy (MOTIE, Korea).

## Author contributions

B.K., J.T., K.K., J.M. and Hyungsuk Lee conceived the project idea. B.K. and J.T. conducted experiments, analyzed the data, and drafted the manuscript. K.K. conducted the experiments and analyzed the data. D.K. conducted a simulation-based analysis. Hyungsoo Lee, S.M., Y.S.P., J.Y., S.L., C.U.L., G.J. and J.L. supported the experiments. J.M. and Hyungsuk Lee supervised the project, directed the research, and contributed to writing the manuscript.

## Competing interests

The authors declare no competing interests.
