## [Peer Review File · Nature Communications]

REVIEWER COMMENTS

Reviewer #1 (Remarks to the Author):

This manuscript presents a protection method for photoelectrochemical water splitting devices using a cryogelated overlayer. The authors synthesized a 1200 μm -thick cryogelated overlayer on a Pt/TiO₂/Sb₂Se₃ photocathode, which remained intact after 210 hours of operation at the range of -12 to -5 mA·cm⁻². The PEC performance preserved approximately 30% of initial photocurrent. Experimental and theoretical analyses well support their hypothesis that trapped bubbles near the top of the cryogelated overlayer act as regulated bubble nucleation sites. This allows gas molecules to be transported to the outer electrolyte solution, reducing mechanical shear stress on the Pt catalyst surface. The stability enhancement achieved by the cryogelated overlayer is encouraging for practical PEC applications. The cryogelation technique also offers promise for large-scale PEC device development due to its fast and simple fabrication process. The manuscript is well-organized with beautifully designed figures. I recommend accepting this manuscript for publication after minor revisions.

On page 3, between lines 12-16, the authors mentioned the negative impact of photo-corrosion on the stability of the photoelectrode. While preventing photo-corrosion is indeed crucial for achieving high performance in PEC devices, the primary focus of this manuscript is on preventing agglomeration and/or peeling off of the Pt HER co-catalyst. The statement made later that the use of a co-catalyst such as Pt reduces the redox overpotential instead of the photo-corrosion reaction may be misleading. This is because, even with the presence of a Pt catalyst, if the protective layer (e.g., TiO₂) is not conformal and complete, the photocathode can still undergo photo-corrosion. It would be beneficial for the authors to discuss how the TiO₂ layer in their electrode composition helps prevent photo-corrosion.

On page 11, between lines 17-20, the authors explained that the gradual photocurrent decay in the long-term test for the 1200 μm -thick sample is due to the accumulation of trapped microbubbles, which reduces the mass transfer of electrolyte and light transmission. Interestingly, the PEC performance can be recovered when mechanically peeling off the cryogelated overlayer. I am curious, whether we could expect the recovery of PEC performance if the trapped bubbles were purged out while still maintaining the presence of the cryogel.

On page 13, between lines 7-11, the authors discussed the problems when operating the PEC device under more practical conditions. Are there any possible solutions?

It would be helpful to offer some comments or insights on the photoanode side: is cryogelated overlayer a good material candidate for protecting photoanode from O₂ bubbles formation?

Reviewer #2 (Remarks to the Author):

The study aims to develop a protection strategy for thin-film PEC devices using a cryo-gelated overlayer to ensure the structural stability of the catalyst for long-term operation. Hydrogen gas trapped in the cryogel protector reduced shear stress at the catalyst surface by providing bubble nucleation sites. Compared to previous work [Nature Energy volume 7, pages 537–547 (2022)], slightly better PEC stability is achieved here by changing the polyacrylamide overlayer fabrication procedure. Overall,

presenting the same idea with slight optimization seems like an insufficient scientific advance to justify another Nature publication. Therefore I believe the MS is more suitable for other journals.

Other points:

- 1) The manuscript is focusing on PEC and the protective characteristics of cryogel. Yet, the chemistry part is completely missing. For example, the characteristics of SAM AMTES, and after the reaction with Glutaraldehyde are missing. How such a formation of a monolayer will influence the activity of the photoelectrode? It would be good if the authors will explain why it is stable after hydrogen evolution for 200 h. The chemical analysis of the electrode after the H₂ evolution will be good to see.
- 2) Degradation of Sb₂Se₃/TiO₂/Pt after just 2.5 h is very quick. Is it true for other systems? Does material or structure play any role?
- 3) How is the UV stability of polyacrylamide?
- 4) Are there other degradation mechanisms plausible?

Reviewer #3 (Remarks to the Author):

1. Micropores can provide sites for bubble nucleation, how to ensure bubble nucleation away from the surface of the optoelectronic device?
2. The pore distribution on the electrode surface should be difficult to achieve uniformity, then the bubble generation at different locations will not be consistent, and it is recommended to supplement the bubble images of multiple locations.
3. Micropores causes bubbles to be less likely to detach and agglomerate, so it may also be detrimental to the electrolyte transfer, will this be detrimental to the efficiency of the optoelectronic device?
4. The authors mentioned that “The overlayer thickness increased the degree of the J_{ph} drop and decreased the recovered J_{ph} value” in page 6. However, it can be seen that the J_{ph} drop degree of CG-1200μm is smaller than CG-800μm and CG-400μm in Fig. 1c.
5. The authors mentioned that “The maximum stress diminished when the overlayer thickness exceeded a critical value (Fig. 2d)” in Page 9. What’s the critical overlayer thickness? And when exceed the critical value, Whether the thicker the overlayer is the better?
6. The mechanical stress caused by nucleation and growth of bubbles may lead to migration and aggregation of the cocatalysts and shorten lifetime of electrodes. The authors mentioned that “We observed a large number of microbubbles at the boundary of the large bubble in the overlayer for CG-1200μm during the long-term operation (Fig. 4c)” in Page 11. Why does the presence of microbubbles not cause migration and aggregation of cocatalysts?
7. How did the authors ensure the uniformity of interconnected porous polyacrylamide cryogelated overlayer on top of Pt/TiO₂/Sb₂Se₃/Au/fluorine-doped tin oxide configured photocathode?
8. In FIG. 1c, why is the time different when the photocurrent density starts to recover after adding different thickness of the cryogelated overlayer?

9. It can be seen from the text that the photocurrent density fluctuation and reduction under the condition of covering 1200 μm cryogelated overlayer are significantly smaller than those under the conditions of 400 μm and 800 μm . So can we conclude that the thicker the cryogelated overlayer, the better?

10. Page 7 Line 4 -Line 10: Compared with the authors' previous work, the device with thick overlayer in this study has the interconnected micropores to help the bubble escape. But the current density of thicker overlayer, like CG-1200 μm , seems lower than the current density of CG-800 μm during long-term operation in Fig. 1d, which also means a higher J_{ph} degradation rate. Which thickness is better for the overall gas production efficiency.

11. Since the reaction temperature is generally above room temperature, does the irradiated light influence the cryogelated overlayer, such as melting or enlarging the pore size.

12. From Supplementary Fig. 7, the pore size on the surface of PEC devices has a large correlation with the ion transport and the contact area of the electrolyte. How the experiment ensures a consistent distribution of contact area between the different PEC devices and the cryogelated overlayers.

Point-by-point response to reviewers' comments

Title: “Stable water splitting using photoelectrodes with a cryogelated overlayer”

Author(s): Byungjun Kang[†], Jaiwan Tan[†], Kyungmin Kim, Donyoung Kang, Hyungsoo Lee, Sunihl Ma, Young Sun Park, Juwon Yun, Soobin Lee, Chan Uk Lee, Gyumin Jang, Jeongyoub Lee, Jooho Moon*, Hyungsuk Lee*

<Reviewer #1>

Remark:

This manuscript presents a protection method for photoelectrochemical water splitting devices using a cryogelated overlayer. The authors synthesized a 1200 μm -thick cryogelated overlayer on a Pt/TiO₂/Sb₂Se₃ photocathode, which remained intact after 210 hours of operation at the range of -12 to -5 mA·cm⁻². The PEC performance preserved approximately 30% of initial photocurrent. Experimental and theoretical analyses well support their hypothesis that trapped bubbles near the top of the cryogelated overlayer act as regulated bubble nucleation sites. This allows gas molecules to be transported to the outer electrolyte solution, reducing mechanical shear stress on the Pt catalyst surface. The stability enhancement achieved by the cryogelated overlayer is encouraging for practical PEC applications. The cryogelation technique also offers promise for large-scale PEC device development due to its fast and simple fabrication process. The manuscript is well-organized with beautifully designed figures. I recommend accepting this manuscript for publication after minor revisions.

Response:

We would like to gratefully thank the reviewer for reviewing and evaluating our work. We believe that the reviewer's comments highly improve the quality of our manuscript. Our response to the reviewer's comments can be found below.

Comment 1:

On page 3, between lines 12-16, the authors mentioned the negative impact of photo-corrosion on the stability of the photoelectrode. While preventing photo-corrosion is indeed crucial for achieving high performance in PEC devices, the primary focus of this manuscript is on preventing agglomeration and/or peeling off of the Pt HER co-catalyst. The statement made later that the use of a co-catalyst such as Pt reduces the redox overpotential instead of the photo-corrosion reaction may be misleading. This is because, even with the presence of a Pt catalyst, if the protective layer (e.g., TiO₂) is not conformal and complete, the photocathode can still undergo photo-corrosion. It would be beneficial for the authors to discuss how the TiO₂ layer in their electrode composition helps prevent photo-corrosion.

Response:

We agree with the reviewer that our previous manuscript might mislead the readers. As the reviewer suggested, we have included the statement regarding the prevention of photocorrosion of the device by incorporating the TiO₂ protection layer in the revised manuscript.

Revision made (colored in blue):

(in Page 3)

However, the lifetime of thin-film photoelectrodes remains insufficient for commercial use because they suffer from severe photocorrosion, which competes with the water splitting reaction at the photoelectrode/electrolyte interface. Use of co-catalysts such as Pt on the photoelectrode surface can drive photogenerated charges towards the favored water-splitting reaction by reducing the redox overpotential instead of the photocorrosion reaction. **Conformal coating of protection layer (e.g., TiO₂) typically extends the device lifetime by physically separating the semiconductor from the corrosive electrolyte. However, we observed that the sluggish kinetics of surface electrochemical reaction (e.g., HER) could lead to the dissolution of TiO₂¹⁰. Despite uniform TiO₂ deposition on semiconductor, agglomeration and detachment of the co-catalyst from the photoelectrodes can accelerate the photocurrent degradation reducing the device lifetime. However, agglomeration and detachment of the co-catalyst from the photoelectrodes**

~~accompanying the photocurrent degradation were frequently observed during the continuous PEC operation, shortening the device lifetime.~~

Comment 2:

On page 11, between lines 17-20, the authors explained that the gradual photocurrent decay in the long-term test for the 1200 μm -thick sample is due to the accumulation of trapped microbubbles, which reduces the mass transfer of electrolyte and light transmission. Interestingly, the PEC performance can be recovered when mechanically peeling off the cryogelated overlayer. I am curious, whether we could expect the recovery of PEC performance if the trapped bubbles were purged out while still maintaining the presence of the cryogel.

Response:

We totally agree with the reviewer's expectation regarding the recovery of the PEC performance by the purge of the trapped bubble. However, it was difficult to manually purge the trapped bubbles without any damage to the cryogel overlayer. For example, manually connecting the purging line to the cryogel ended up with the burst of a trapped bubble followed by the delamination of the cryogel. Instead, we hypothesized that the bubbles can be removed by the spontaneous diffusion of gas molecules when we periodically stop the PEC operation, *i.e.*, no more gas production. This is the reason why we experimentally simulated the daily cycle test, consisting of a 12 h-day cycle for gas production and a 12 h-night cycle for no gas production (**Supplementary Note 3**). We found that trapped bubbles in the cryogel were removed during the night cycle, albeit the performance of the device was not recovered. Not only the bubble extraction from cryogel overlayer but also maintaining a good quality interface of photoelectrodes during the daily cycle highly mattered for the recovery of the performance.

Please see more details in the next comment. The revision made for this comment can be found after our response to your Comment 3.

Comment 3:

On page 13, between lines 7-11, the authors discussed the problems when operating the PEC device under more practical conditions. Are there any possible solutions?

Response:

We appreciate the reviewer's comment regarding the problems during the PEC operation in practical conditions. The main problem was that the photocurrent did not recover after the first night cycle, despite the bubble removal from the cryogel overlayer. It might be due to the interface oxidation of the device, as evidenced by the sudden change of open circuit potential from the negative to the positive value (**Supplementary Note Fig. 4 of the original manuscript**). Maintaining the constant potential during the night cycle could be one plausible solution. Another solution is shorting the night cycle duration (light-off cycle) to prevent this sudden potential change. As shown in **Fig. R1**, the *CG-1200 μ m* operated over 400 h with 3 h-on/1 h-off illumination cycle. For the first few periodic cycles, the J_{ph} was almost recovered after the light-off cycle. We still observed gradual degradation because the bubbles were not fully removed during the light-off cycle. Although 3 h-on/1 h-off is not the practical condition of the daily cycle, this result clearly demonstrates that the PEC performance can be recovered by removing trapped bubbles.

In the revised manuscript, we clearly described our motivation for the on/off cycle testing in terms of removing the trapped bubbles (related to comment 2) and plausible solutions for practical operation. **Fig. R1** was included as **Supplementary Fig. 17** in the revised manuscript.

Fig. R1. On/off cycle operation of a Sb₂Se₃ photocathode covered with the cryogelated overlayer with interconnected micropores. (a) The photocurrent density-time profile of *CG-1200*μm operated by on and off cycles during 20 h. The duration of the on and off cycles was 3 h and 1 h, respectively. (b) Pictures of the bubbles in *CG-1200*μm after the ‘on cycle’ and ‘off cycle’. (c) The photocurrent density-time profile of *CG-1200*μm operated by on and off cycles for 480 h.

Revision made (colored in blue):

(in Page 14)

Since the J_{ph} of *CG-1200*μm was reduced without structural damage of the device, we hypothesized that the J_{ph} can be recovered when the trapped bubbles are effectively removed. It is assumed that bubbles can be removed by the spontaneous diffusion of gas molecules when we periodically stop the PEC operation for the cessation of gas production. We performed a chronoamperometry test with light chopping, repeating 3 h of light-on and 1 h of light-off cycle (**Supplementary Fig. 17**). For the first few cycles, the J_{ph} was almost recovered after each light-off cycle albeit it gradually decreased because the bubbles were not completely removed. This result clearly demonstrates that the PEC performance can be recovered by removing trapped bubbles. To test the device in a more practical condition, ~~we performed a chronoamperometry test with on/off illumination cycles,~~

~~simulating~~ we simulated the daily day and night cycle (see details in Note S3). We observed that microbubbles trapped in the overlayer during the day cycle can escape during the dark cycle.

(in Page 28 in Supplementary Information)

Nevertheless, it was observed that the microbubbles accumulated in the micropores could be removed by ‘diffusion of gas molecule’ through the pores during the night cycle when the additional gas supply is inhibited (**Supplementary Note Fig. 4c**). Thus, J_{ph} was maintained at $\sim 4 \text{ mA}^{-2}$ over 10 day/night cycles. As evidenced in **Supplementary Fig. 17**, preventing the instant potential change during the night cycle allows stable operation; therefore, maintaining the constant potential during the night cycle could be one plausible solution.

Comment 4:

It would be helpful to offer some comments or insights on the photoanode side: is cryogelated overlayer a good material candidate for protecting photoanode from O₂ bubbles formation?

Response:

We appreciate the reviewer’s comment regarding the application of cryogelated overlayer for photoanode. The main function of our cryogel overlayer is to provide a nucleation site for bubbles “away from” the device surface, which is a beneficial strategy to protect the surface co-catalyst from the physical shear stress-induced structural degradation (*e.g.*, agglomeration and detachment). Therefore, we expect our cryogel overlayer protection would also be beneficial for the photoanode side when the co-catalyst is chemically stable in the electrolyte.

The pH of the electrolyte surrounding the hydrogel influences the swelling of the gel which can alter the pore size and thickness of the gel. By incorporating the internal porous structure and dimension tailored specifically to the operation conditions of the device, the cryogel overlayer has a potential to enhance the structural stability of the photoelectrochemical devices, including photoanodes decorated with the chemically stable co-catalysts.

In the revised manuscript, we have included the statement regarding further considerations when applying the cryogel overlayer for different types of PEC devices such as photoanode.

Revision made (colored in blue):

(in Page 14)

We expect that our cryogel overlayer protection would also be beneficial for the photoanode side when the co-catalyst is chemically stable in the electrolyte. The pH of the electrolyte may affect the swelling of the hydrogel, leading to changes in the structural properties of the gel, such as pore size and thickness. By incorporating the internal porous structure and dimension tailored specifically to the operation conditions of the device, the cryogel overlayer would have the potential to enhance the structural stability of the PEC devices, including photoanodes decorated with chemically stable co-catalysts.

<Reviewer #2>

Remark:

The study aims to develop a protection strategy for thin-film PEC devices using a cryo-gelated overlayer to ensure the structural stability of the catalyst for long-term operation. Hydrogen gas trapped in the cryogel protector reduced shear stress at the catalyst surface by providing bubble nucleation sites. Compared to previous work [Nature Energy volume 7, pages 537–547 (2022)], slightly better PEC stability is achieved here by changing the polyacrylamide overlayer fabrication procedure. Overall, presenting the same idea with slight optimization seems like an insufficient scientific advance to justify another Nature publication. Therefore I believe the MS is more suitable for other journals.

Response:

We appreciate the reviewer's comment of concern regarding the scientific advance in our manuscript. We would like to emphasize that the motivation of the manuscript was to acquire a scientific understanding of the protection mechanism for overlayers with porous structure as well as extending the device operation duration through structure optimization.

In the previous study (*Nature Energy* 7, 537–547 (2022)), the agglomeration and detachment of Pt catalyst, caused by the shear stress from the bubbles, were diminished by coating the PEC device with nanoporous hydrogel. The structural stability of the catalyst layer and the PEC device might be attributed to the immobilization of catalyst particles by the spatial confinement or reduction of shear stress on the catalyst by regulating the bubble dynamics.

In this study, we intended to understand the mechanism of how the porous protector improves the structure stability of the device. The effect of the direct spatial confinement of Pt catalyst on the device stability was investigated by using cryogel overlayers engineered to have micropores with sizes of tens of micrometers which is much larger compared to catalyst nanoparticles. When the microporous cryogel overlayer was coated on the device, the photoelectrochemical performance of the device and the structure of the Pt catalyst remained intact after the 210 h of the PEC operation. This result indicated that the direct spatial confinement on the catalyst nanoparticles by the

hydrogel is not critical in stabilizing the catalyst layer. The effect of hydrogen bubbles on the device stability was investigated by regulating their trapping using the engineered porous protector. We revealed that the surface structure of the device was significantly degraded when no large bubble trapping occurred by the additional macropores formed at the outer surface of the cryogel protector (**Fig. 5b of the original manuscript**). These results indicate that the bubbles trapped in the protector might inhibit the bubble nucleation at the device surface, prevent the shear stress exerted on the device surface, and reduce the structural degradation of the Pt catalyst consequently.

To figure out the mechanism further, this manuscript provides a theoretical and numerical model describing the expansion and escape dynamics of the bubbles for a PEC device with a porous hydrogel protector. Using those models, we revealed how the mechanical fracture and softening of the protector caused by the bubbles could play a role in determining the structural and functional stability of the PEC devices.

To the best of our knowledge, this study is the first to elucidate the mechanism of stability of PEC devices enhanced by incorporating a porous overlayer protector. The scientific advances in this study can be applied to develop semi-permanent electro- and photoelectro-chemical devices.

We revised the abstract, introduction, results, and conclusion to state the scientific contributions of our study more clearly.

Revision made (colored in blue):

(in Page 2; in Abstract)

However, solar water splitting systems suffer from short lifetimes due to catalyst instability, which might be attributed to the mechanical stress produced by hydrogen bubbles, preventing them from being used commercially. **The recent study found that the nanoporous hydrogel could prevent the structural degradation of the PEC devices. In this study, we investigate the protection mechanism of the hydrogel-based overlayer by engineering its porous structure using the cryogelation technique. we developed a protection strategy for thin film PEC devices using a cryogelated overlayer to ensure the structural stability of the catalyst for long-term operation.** Tests for cryogel overlayers with varied microstructures reveal that the hydrogen gas trapped in the cryogel protector

reduce shear stress at the catalyst surface by providing bubble nucleation sites. The cryogelated overlayer effectively preserves the uniformly distributed platinum catalyst particles on the device surface for over 200 h. Our ~~finding strategy~~ can help establish semi-permanent photoelectrochemical devices to realize a carbon-free society.

(in Page 4; in Introduction)

We hypothesized that the structural stability of the device could be improved by ~~regulating the mechanical stress applied on the co-catalyst during the gas evolution reaction. modulating the products of the gas evolution reaction in the PEC devices.~~ Once the hydrogen or oxygen molecules produced at the co-catalysts exceed the critical concentration required to overcome the free-energy barrier of bubble nucleation, they follow the nucleation and growth theory along the device surface until they detach by increasing buoyancy¹⁵. The surface-growing bubbles apply mechanical force to the device surface¹⁶, which may cause migration and aggregation of the co-catalysts^{17,18}. The surface nanostructuring, coating of aerophobic polymer, or addition of surfactant to the electrolyte solution could reduce the critical size of bubble required for the bubble detachment¹⁹⁻²⁴. However, a shear stress is still produced on the catalyst surface by bubbles in a periodic manner through repeated cycles of bubble nucleation, expansion, and detachment.

The recent study suggested that the coating of the nanoporous hydrogel on the PEC device could significantly enhance the structural stability of the catalyst and PEC devices²⁵. The structural stability of the catalyst layer might be attributed to the immobilization of catalyst particles by the spatial confinement of hydrogel or reduction of shear stress on the catalyst by regulating the bubble dynamics. In this study, we sought to understand the mechanism of how the porous protector improves the structure stability of the device by incorporating overlayer having various types of porous structures. ~~A few attempts were proposed to regulate the bubble cycle reducing the shear stress at the electrode surface. When bubbles were pre-trapped at micromachined superhydrophobic pillar placed away from the device, they absorb produced gas molecules, which prevented bubble nucleation at the surface²⁶. However, the pre-trapped bubbles were easily detached from the pillar due to the increased buoyancy as they expanded. We hypothesized that a 3-dimensional microporous hydrogel-based matrix can retain bubbles at a location away from the device surface while minimizing the deteriorating effect of the bubble on the PEC performance.~~

~~In this study,~~ We employ a cryogelation technique to fabricate a microporous hydrogel overlayer because it is advantageous over other techniques such as porogen/leaching and gas-foaming ones in terms of fabrication speed and simplicity²⁶. Experiments show that, during the PEC operation, gas bubbles are nucleated and trapped in the cryogel to provide nucleation sites of bubbles for transporting gas molecules to the outer electrolyte solution. Porous structure of the cryogel overlayer contributes to reduce the mechanical shear stress produced by nucleated bubbles at the device surface preserving the surface structure of the Pt/TiO₂/Sb₂Se₃ photocathode even after 210 h operation. Theoretical and numerical analyses reveal how the structural and functional stabilities of the PEC devices are determined by mechanical fracture and softening of the protector occurred by bubbles. The scientific advances in this study can be applied to develop semi-permanent electro- and photoelectro-chemical devices.

(in Page 6; in Results)

The internal structure of the cryogelated overlayer was visualized by three-dimensional confocal fluorescence imaging of 200 nm fluorescent nanoparticles attached on the polymer (**Fig. 1b and Supplementary Fig. 1**). The average pore size was approximately 20 μm which is much larger compared to catalyst nanoparticles. Thus, the effect of spatial confinement on the catalyst by the cryogel would be smaller than that by a regular hydrogel with nanopores used in the previous study²⁵.

(in Page 6; in Results)

~~To determine the effective dimension of the cryogel as an overlayer of PEC devices,~~ To investigate the effect of the microporous cryogel overlayer on the performance and the structural stability of the PEC devices, we performed chronoamperometry at 0 V versus a reversible hydrogen electrode (V_{RHE}) under 1-sun illumination on various Sb₂Se₃ photocathodes covered with cryogel overlayers having thicknesses of 0, 100, 200, 400, 800, and 1200 μm , denoted as *no cryogel*, *CG-100 μm* , *CG-200 μm* , *CG-400 μm* , *CG-800 μm* , and *CG-1200 μm* , respectively.

(in Page 8; in Results)

Scanning transmission electron microscopy (STEM) and energy-dispersive X-ray spectroscopy (EDS) analyses of the cross-sectioned Sb_2Se_3 photocathode after peeling off the 1200 μm -thick overlayer showed that uniform Pt nanoparticles were clearly observed on the $\text{TiO}_2/\text{Sb}_2\text{Se}_3$ nanorods even after photocurrent degradation (**Fig. 1g**). This result suggested that the direct spatial confinement of catalyst nanoparticles by the nanoporous overlayer²⁵ might not play a critical role in stabilizing the catalysts.

(in Page 15; in Conclusion)

In this study, we investigated the protection mechanism for overlayers with porous structure by utilizing engineered cryogel overlayers. ~~In this study,~~ We found that bubbles trapped near the top of the cryogel during the early PEC operation function as regulated bubble nucleation sites, providing a temporal pathway for transporting gas molecules to the outer electrolyte solution. As a result, the mechanical shear stress produced by bubbles at the device surface can be reduced. Employing the 1200 μm -thick cryogelated overlayer, the structure of the $\text{Pt}/\text{TiO}_2/\text{Sb}_2\text{Se}_3$ photocathode remained intact ~~even~~ after 210 h operation, exhibiting approximately 30% J_{ph}/J_0 .

We sincerely appreciate the reviewers for thoughtful comments on our manuscript. Our point-by-point responses to other comments raised by the reviewer can be found below.

Comment 1:

The manuscript is focusing on PEC and the protective characteristics of cryogel. Yet, the chemistry part is completely missing. For example, the characteristics of SAM AMTES, and after the reaction with Glutaraldehyde are missing. How such a formation of a monolayer will influence the activity of the photoelectrode? It would be good if the authors will explain why it is stable after hydrogen evolution for 200 h. The chemical analysis of the electrode after the H₂ evolution will be good to see.

Response:

We appreciate an important comment regarding the chemical analysis of photoelectrodes. We are sorry that the chemical analysis of the device was not sufficient in the original manuscript.

The self-assembled monolayer (SAM) of (3-aminopropyl)triethoxysilane (APTES)-glutaraldehyde has been widely utilized to immobilize the polyacrylamide hydrogel or proteins on the TiO₂ or SiO₂ substrate (*Journal of Biomedical Materials Research Part A*. **90A**, 35-45 (2009); *Journal of Physics: Condensed Matter* **22**, 194116 (2010); *Biosensors* **13**, 36 (2023)). Silanol group of hydrolyzed APTES can react with hydroxyl group of the substrate formed by the NaOH treatment. The amine group of APTES then reacts with aldehyde group of glutaraldehyde. During the gelation of the polyacrylamide hydrogel, the polymer network of polyacrylamide can be combined with another end of glutaraldehyde.

A typical method of chemical characterization on organic layers is FT-IR analysis. Well-controlled ATR configuration might be one possible way, but normally it is difficult to detect signals from monolayers deposited on nanostructured films. As described above, because the SAM of APTES is already well-studied and characterized, we think it is not necessary to provide chemical analysis on the monolayer for our hydrogel system. We still can confirm the existence of the SAM because the cryogel does not withstand on the device without the SAM treatment. According to the

additional LSV test, the SAM of APTES and glutaraldehyde did not significantly influence the PEC performance of the device without the cryogel overlayer (**Fig. R2**).

Fig. R2. The effect of self-assembled monolayer (SAM) treatment on the PEC performance. The current density J_{ph} -potential curve of Pt/TiO₂/Sb₂Se₃ devices w/o treatment (as-prepared), after the APTES treatment step, and after the glutaraldehyde treatment step. The SAM treatment did not significantly alter the PEC performance.

We further compared the XRD spectra before and after the PEC operation for over 200 h. The crystallinity and orientation of Sb₂Se₃ were maintained, and no secondary phases were observed even after the PEC operation over 200 h (**Fig. R3**).

Fig. R3. X-ray diffraction (XRD) spectra of Pt/TiO₂/Sb₂Se₃ device before and after the stability test. The crystallinity and orientation of Sb₂Se₃ were maintained, and no secondary phases were observed after the PEC operation over 200 h.

In the revised manuscript, we included detailed descriptions of the SAM treatment with the LSV and XRD data on Sb₂Se₃ photoelectrodes before and after the stability test. **Fig. R2** and **Fig. R3** were included as **Supplementary Fig. 18 and 8**, respectively.

Revision made (colored in blue):

(in Page 8)

In addition, the crystallinity and orientation of Sb₂Se₃ were maintained, and no secondary phases were observed even after the PEC operation over 200 h (**Supplementary Fig. 8**). These observations raise two important questions: what is the origin of the gradual J_{ph} degradation and what is the mechanism over which the cryogelated overlayer protects the surface microstructure of the device?

(in Page 18)

The self-assembled monolayer (SAM) of APTES-glutaraldehyde is widely used to immobilize the polyacrylamide hydrogel or proteins on the TiO₂ or SiO₂ substrate³⁷⁻³⁹. The silanol group of hydrolyzed APTES reacts with the hydroxyl group of the substrate formed by the NaOH treatment. The amine group of APTES then reacts with the aldehyde group of glutaraldehyde. During the gelation of the polyacrylamide hydrogel, the polymer network of polyacrylamide can be combined with another end of glutaraldehyde. However, this monolayer did not significantly influence the PEC performance without the cryogel overlayer (**Supplementary Fig. 18**). Without this SAM treatment step, the cryogel was easily detached from the device.

Comment 2:

Degradation of Sb₂Se₃/TiO₂/Pt after just 2.5 h is very quick. Is it true for other systems? Does material or structure play any role?

Response:

We appreciate the reviewer's comment about the degradation characteristics of Sb₂Se₃/TiO₂/Pt. As the reviewer commented, the structural degradation and lifetime of the PEC devices are dependent on the constituent material and structure of the devices. The degradation mechanism of Sb₂Se₃/TiO₂/Pt device was demonstrated in our previous study (*Advanced Energy Materials* **9**, 1900179 (2019)). After ~ 1 h of stable operation, the photocurrent linearly decreased accompanying the reductive dissolution of TiO₂ layer. When Sb₂Se₃ absorbing layer was directly exposed to the electrolyte due to the dissolution of the TiO₂ layer, the photocurrent began to exponentially decrease leading to severe corrosion of inner layer. The rapid photocurrent degradation after 2.5 h is due to the exponential decrease accompanying the corrosion of Sb₂Se₃. The specific degradation process might depend on the material and structures. We expect that the photocathodes with TiO₂ deposition and Pt catalyst decoration will show similar degradation behaviors. The lifetime of the Pt/TiO₂/Sb₂Se₃ was comparable with other types of low-cost thin-film photocathodes (**Table R1**).

Table R1. Device components and operation duration photocathodes composed of Pt catalyst, TiO₂ layer, and low-cost thin-film light absorber.

No.	Device components	Duration	Reference
1	Pt/TiO ₂ /Sb ₂ Se ₃	2 h	Adv. Energy Mater. 8, 1702888 (2018)
2	Pt/TiO ₂ /AZO/Cu ₂ O	20 min	Nat. Mater. 10, 456–461 (2011)
3	Pt/TiO ₂ /Ga ₂ O ₃ /Cu ₂ O	2 h	Energy Environ. Sci. 8, 1493–1500 (2015)
4	Pt/TiO ₂ /CdS/CuO	30 min	Chem. Mater. 29, 1735–1743 (2017)
5	Pt/TiO ₂ /CdS/CZTS	1 h	ACS Energy Lett. 1, 1127–1136 (2016)

In the revised manuscript, we added the statement regarding the operation duration of other types of thin-film photocathodes and included **Table R1** as **Supplementary Table 1**.

Revision made (colored in blue):

(in Page 7)

This indicates that the thick overlayer significantly extended the lifetime of the PEC device compared to that of the reference device, and *no cryogel* showed full J_{ph} degradation within 5 h (inset of Fig. 1d) which was similar to the operation durations for other low-cost thin-film photocathodes (Supplementary Table 1).

Comment 3:

How is the UV stability of polyacrylamide?

Response:

We appreciate the reviewer's comment regarding the UV stability of the polyacrylamide. According to the previous studies (*Polymer* **44**, 1331-1337 (2003); *Journal of Chromatographic Science* **37**, 486-494 (1999)), the polyacrylamide hydrogel was not significantly degraded and thus highly stable under the illumination of UV light with the wavelength of 254 nm and sunlight for more than ten days. Also, we observed that the polyacrylamide hydrogel was structurally stable inside the 0.1 M H₂SO₄ electrolyte under light illumination for 1000 h (Fig. R4).

Fig. R4. Optical microscopic images of the polyacrylamide hydrogel after immersion in a 0.1 M H₂SO₄ solution for 0, 24, and 1000 h. Red dotted lines represent the boundaries of the hydrogel.

In the revised manuscript, we have included the statement regarding the stability of polyacrylamide hydrogel under light illumination.

Revision made (colored in blue):

(in Page 6)

The polymeric part of the cryogel overlayer is polyacrylamide hydrogel which is nanoporous, poroelastic, water-permeable, hydrophilic, and aerophobic^{25,29,30}. The polyacrylamide hydrogel was tested to be stable under light illumination^{31,32}.

Comment 4:

Are there other degradation mechanisms plausible?

Response:

We appreciate the reviewer's comment regarding a plausible degradation mechanism. Two major factors of photoelectrode degradation are the photocorrosion of semiconductor and the degradation of surface co-catalyst. To prevent the photocorrosion, various protection strategies, including TiO₂ deposition by atomic layer deposition (ALD), have been employed to physically separate the semiconductor from the corrosive electrolyte (*Advanced Energy Materials* **9**, 1900179 (2019)). Although TiO₂ protection improved the stability of photocathodes to some extent as compared with their unprotected counterparts, a gradual photocurrent degradation was still unavoidable. We recently discovered that these protective layers cannot be permanently incorporated; for example, surface-accumulated photoelectrons induce self-reduction and dissolution of TiO₂, thereby gradually exposing the inner semiconductor to the corrosive electrolyte. Maintaining the structural stability of the Pt catalyst is significant to prevent surface accumulation of photoelectrons and dissolution of TiO₂.

While preventing photocorrosion is also crucial for achieving stable PEC devices, the primary focus of our manuscript is on maintaining the structural stability of the Pt co-catalyst. Since this description might mislead the readers, we have clearly modified the role of the TiO₂ protection layer in terms of inhibiting the photocorrosion in the Introduction of the revised manuscript.

Revision made (colored in blue):

(in Page 3)

However, the lifetime of thin-film photoelectrodes remains insufficient for commercial use because they suffer from severe photocorrosion, which competes with the water splitting reaction at the photoelectrode/electrolyte interface. Use of co-catalysts such as Pt on the photoelectrode surface can drive photogenerated charges towards the favored water-splitting reaction by reducing the redox overpotential instead of the photocorrosion reaction. Conformal coating of protection layer (e.g., TiO₂) typically extends the device lifetime by physically separating the semiconductor from the corrosive electrolyte. However, we observed that the sluggish kinetics of surface electrochemical reaction (e.g., HER) could lead to the dissolution of TiO₂¹⁰. Despite uniform TiO₂ deposition on semiconductor, agglomeration and detachment of the co-catalyst from the photoelectrodes can accelerate the photocurrent degradation reducing the device lifetime. ~~However, agglomeration and detachment of the co-catalyst from the photoelectrodes accompanying the photocurrent degradation were frequently observed during the continuous PEC operation, shortening the device lifetime.~~

<Reviewer #3>

Response:

We would express our gratitude to the reviewer for the insightful comments on our manuscript. We firmly believe that the comments raised by the reviewer significantly improved the quality of our manuscript. The point-by-point responses to the reviewer's comments can be found below.

Comment 1:

Micropores can provide sites for bubble nucleation, how to ensure bubble nucleation away from the surface of the optoelectronic device?

Response:

We appreciate the reviewer's constructive comment regarding the location of the bubble nucleation. According to the additional experiment, the double layer capacitance (E_{dl}) of the device, indicative of the electrochemical surface area, was similar when measured before and after the bubble formation (**Fig. R5**). This result indicates that the bubbles were nucleated at a site away from the device surface.

Fig. R5. The electrochemical surface area (ECSA) analysis using cyclic voltammetry before and after the formation of the initial bubble. The double layer capacitance E_{dl} of $CG-400\mu m$ before and after the bubble formation was similar. The ECSA can be calculated by dividing E_{dl} by a specific capacitance determined by the electrodes.

In the revised manuscript, we included the statement regarding the effect of the formation of the bubbles in the cryogel on the ECSA of the device. **Fig. R5** was included as **Supplementary Fig. 11** in the revised manuscript.

Revision made (colored in blue):

(in Page 9)

The large bubble might lead to the reduction of J_{ph} , as observed within 0.5 h after operation, via hindering the electrolyte transport and light transmittance (**Fig. 1c**). *The electrochemical surface area measured before and after the bubble formation was similar (Supplementary Fig. 11), suggesting that the bubble was nucleated at a site away from the device surface.* The large trapped bubbles were further expanded by the continuous gas production, deforming the overlayer.

Comment 2:

The pore distribution on the electrode surface should be difficult to achieve uniformity, then the bubble generation at different locations will not be consistent, and it is recommended to supplement the bubble images of multiple locations.

Response:

We appreciate the reviewer's comment regarding the location-dependent bubble formation in the PEC device and revised the manuscript to include bubble images at multiple locations, as the reviewer recommended. As shown in **Fig. R6**, small bubbles with different shapes and positions were observed at the early stages of the PEC operation, which might be attributed to the varied shape and dimensions of pores (**Fig. 1b and Supplementary Note Fig. 2c of the original manuscript**). Those bubbles were transformed into large bubbles through expansion and merging processes at the later stage of operation.

Fig. R6. Time-lapse images of the formation and merge of microbubbles in $CG-400\mu m$ during the early stages of the PEC operation. Scale bar represents 0.5 mm.

In the revised manuscript, we have included the statement regarding the inhomogeneity in the porous structure of the cryogel. **Fig. R6** was included as **Supplementary Fig. 10** in the revised manuscript.

Revision made (colored in blue):

(in Page 9)

To determine the bubble dynamics, including nucleation, expansion, and escape during the initial J_{ph} drop recovery and the following J_{ph} fluctuation, we visualized bubbles in the cryogelated overlayer *in situ* for $CG-800\mu m$, which showed a large initial J_{ph} drop recovery (**Fig. 2a**). At the early stages of the PEC operation, small gas bubbles with different shapes and positions were observed (**Supplementary Fig. 10**) due to the heterogeneity in shape and dimension of pores (**Fig. 1b**). Those bubbles were then merged into large bubbles at the later stage of operation. The large bubble might lead to the reduction of J_{ph} , as observed within 0.5 h after operation, via hindering the electrolyte transport and light transmittance (**Fig. 1c**).

Comment 3:

Micropores causes bubbles to be less likely to detach and agglomerate, so it may also be detrimental to the electrolyte transfer, will this be detrimental to the efficiency of the optoelectronic device?

Response:

We agree with the reviewer's comment that the trapping of bubbles in the micropores is detrimental to the electrolyte transfer. As a result, the bubbles trapped in the cryogel micropores led to the loss of the device efficiency indicated by the reduction of the photocurrent. We have included the statement about the reduction of the device performance by the bubble trapped in the microporous cryogel overlayer more clearly in the revised manuscript.

Revision made (colored in blue):

(in Page 13)

The accumulation of trapped microbubbles in the overlayer **by its microporous structure (Fig. 4d)** can hinder not only the mass transfer of electrolytes but also light transmission onto the device surface, leading to a gradual reduction in the photocurrent **of the device**.

Comment 4:

The authors mentioned that "The overlayer thickness increased the degree of the J_{ph} drop and decreased the recovered J_{ph} value" in page 6. However, it can be seen that the J_{ph} drop degree of CG-1200 μm is smaller than CG-800 μm and CG-400 μm in Fig. 1c.

Response:

We appreciate the reviewer's comment regarding the degree of J_{ph} drop of CG-400 μm , CG-800 μm , and CG-1200 μm . We apologize for the confusing statement about the degree of J_{ph} drop. The degree of J_{ph} drop was determined as the photocurrent density at the time just before the occurrence

of the photocurrent recovery divided by the initial photocurrent density. The degree of J_{ph} drop of $CG-1200\mu m$ was higher than that of $CG-400\mu m$ and similar to that of $CG-800\mu m$ (**Fig. R7**).

Fig. R7. Degree of current drop characterized by current density at the timepoint right before the recovery of current density divided by the initial current density for $CG-400\mu m$, $CG-800\mu m$, and $CG-1200\mu m$.

We agree with reviewer and we removed the confusing statement regarding the effect of the overlayer thickness on the degree of J_{ph} drop in the revised manuscript.

Revision made (colored in blue):

(in Page 6)

For $CG-400\mu m$, J_{ph} decreased from 16 to 7 mA cm⁻² during the first 0.1 h, followed by a partial recovery to 11 mA cm⁻². Interestingly, after the initial J_{ph} drop and recovery, the recovered J_{ph} was maintained with small and regular fluctuations. ~~The overlayer thickness increased the degree of the J_{ph} drop and decreased the recovered J_{ph} value.~~ $CG-800\mu m$ and $CG-1200\mu m$ exhibited a large initial J_{ph} drop recovery within 0.5 h, followed by a stable J_{ph} value with slight fluctuations.

Comment 5:

The authors mentioned that “The maximum stress diminished when the overlayer thickness exceeded a critical value (Fig. 2d)” in Page 9. What’s the critical overlayer thickness? And when exceed the critical value, Whether the thicker the overlayer is the better?

Response:

We appreciate the reviewer’s comment regarding the critical thickness overlayer in **Fig. 2d** of the original manuscript. We are sorry that the sentence was not clearly described regarding the critical value.

When the relative bubble size equals 4, and a normalized overlayer thickness is higher than 5, the stress produced at the interface between the bubble and cryogel can be decreased (**Fig. 2d of the original manuscript**), and the probability of causing fracture can be reduced. However, at the same time, a probability of pore enlargement, which is essential for bubble escape and corresponding photocurrent recovery, can also be diminished.

In the revised manuscript, we removed the statement regarding the critical thickness and included the statement regarding the effect of the higher overlayer thickness on the fracture of the overlayer.

Revision made (colored in blue):

(in Page 10)

The increase in the maximum stress proportional to the overlayer thickness was more significant at higher bubble expansion ratios (**Fig. 2d**). ~~The maximum stress diminished when the overlayer thickness exceeded a critical value (Fig. 2d).~~ For the relative bubble size equal to 4, the maximum stress slightly decreased as the normalized thickness exceeded 5. The high mechanical stress produced, especially at the bubble edge, could increase the probability of fractures enlarging the local porous area, which is more significant at a thick overlayer (**Fig. 2e**)^{33,34}.

Comment 6:

The mechanical stress caused by nucleation and growth of bubbles may lead to migration and aggregation of the cocatalysts and shorten lifetime of electrodes. The authors mentioned that “We observed a large number of microbubbles at the boundary of the large bubble in the overlayer for CG-1200 μ m during the long-term operation (Fig. 4c)” in Page 11. Why does the presence of microbubbles not cause migration and aggregation of cocatalysts?

Response:

We appreciate the reviewer’s comment regarding the effect of the microbubbles on the structure of the co-catalyst of the devices.

The expansion and contraction of the large bubbles trapped in the cryogel can occur the mechanical deformation of the surrounding cryogel producing the shear stress on the device surface (**Supplementary Fig. 11b of the original manuscript**) and the structural damage of CG-400 μ m (**Fig. 4b of the original manuscript**). In contrast, microbubbles trapped in the cryogel micropores on the CG-1200 μ m, did not seem to damage the structure of Pt co-catalysts (**Fig. 1g of the original manuscript**). Due to their smaller dimension of microbubbles, the mechanical interaction with the cryogel during the bubble dynamics and consequential shear stress on the device surface would be less than large bubbles.

We have included the statement regarding the shear stress by the microbubbles in the revised manuscript.

Revision made (colored in blue):

(in Page 12)

We observed a large number of microbubbles at the boundary of the large bubble in the overlayer for CG-1200 μ m during the long-term operation (**Fig. 4c**). **In contrast to large bubbles, microbubbles in the cryogel did not seem to damage the structure of Pt catalyst on the device surface (Fig. 1g).** We speculated that the mechanical interaction with the cryogel occurred during bubble dynamics and consequential shear stress on the device surface would be less for microbubbles due to their smaller dimension.

Comment 7:

How did the authors ensure the uniformity of interconnected porous polyacrylamide cryogelated overlayer on top of Pt/TiO₂/Sb₂Se₃/Au/fluorine-doped tin oxide configured photocathode?

Response:

We appreciate the reviewer's comments regarding the porosity uniformity of the cryogel overlayer. It was reported that the pore structure of cryogel depends on the temperature and its gradient of the pre-gel solution (*Chemical Engineering Science* **61**, 6701-6708 (2006), *Advanced Materials* **33**, 2100091 (2021)). To be consistent, we prepared the sample in the same location with a constant temperature and minimal temperature gradient in the freezer.

In the revised manuscript, we have included the statement regarding the fabrication condition of the cryogel to ensure the uniformity of the porous cryogelated overlayer.

Revision made (colored in blue):

(in Page 19)

1 ml of pre-gel solution was prepared by mixing 888 μ l of DI water, 65.2 μ l of 100% AAM, 40.8 μ l of 2% BAAM, 5 μ l of 10% AP, and 1 μ l of TEMED. **The cryogelation process was conducted at the same location with a minimal temperature gradient in a freezer.** After cryogelation in a – 20 °C freezer in the laboratory for 12, 24, 48, or 72 h, the rubber mold and cover glass were carefully removed from the cryogelated overlayer.

Comment 8:

In FIG. 1c, why is the time different when the photocurrent density starts to recover after adding different thickness of the cryogelated overlayer?

Response:

We appreciate the reviewer's comment regarding the timepoint of the recovery of the photocurrent density in the PEC devices with cryogel overlayers having varied thicknesses. Initiation and propagation of fracture in the overlayer is stochastic, and it depends on the overlayer thickness, resulting in the difference in the timepoints of the photocurrent recovery for different thicknesses (**Fig. 1c of the original manuscript**).

In the revised manuscript, we included the statement regarding the different time points of the recovery of photocurrent density shown in **Fig. 1c**.

Revision made (colored in blue):

(in Page 10)

The high mechanical stress produced, especially at the bubble edge, could increase the probability of fractures enlarging the local porous area, which is more significant at a thick overlayer (**Fig. 2e**)^{33,34}. *Initiation and propagation of fracture in the overlayer is stochastic, and it depends on the overlayer thickness, resulting in the difference in the timepoints of the photocurrent recovery for different thicknesses (Fig. 1c).*

Comment 9:

It can be seen from the text that the photocurrent density fluctuation and reduction under the condition of covering 1200 μm cryogelated overlayer are significantly smaller than those under the conditions of 400 μm and 800 μm . So can we conclude that the thicker the cryogelated overlayer, the better?

Response:

We appreciate the reviewer's comment regarding the thickness of the cryogel overlayer. The thicker cryogel with small current fluctuation was more susceptible to protecting the device surface structure. However, as the thickness increases, there is a higher probability of the bubble trapping in the cryogel, which can lead to diminished electrolyte transfer and consequential reduction of photocurrent density of the device. Thus, it is required to optimize the thickness of the cryogel by considering the structural stability and photocurrent density.

We have included the discussion regarding the selection of the optimal thickness of the cryogel in the revised manuscript.

Revision made (colored in blue):

(in Page 13)

Our results showed that the thicker cryogel overlayer could increase the structural stability while reducing the current density of the device. Thus, to obtain the optimal thickness of cryogel requires consideration in terms of lifetime and photocurrent density.

Comment 10:

Page 7 Line 4 -Line 10: Compared with the authors' previous work, the device with thick overlayer in this study has the interconnected micropores to help the bubble escape. But the current density of thicker overlayer, like CG-1200 μm , seems lower than the current density of CG-800 μm during long-term operation in Fig. 1d, which also means a higher J_{ph} degradation rate. Which thickness is better for the overall gas production efficiency.

Response:

We appreciate the reviewer's comment regarding the bubble escape through the interconnected pores of the cryogel and the overall gas production efficiency of the devices.

The cryogel overlayer with interconnected micropores could help the bubble escape compared to the nanoporous hydrogel utilized in our previous study (*Nature Energy* 7, 537–547, 2022), but did not completely prevent the trapping of bubbles in the overlayer. Some bubbles were still trapped in the micropores at the region away from the device surface, reducing the current density of the devices.

In a thicker cryogel, more bubbles could be trapped in the pores. As a result, the current density after the initial J_{ph} drop recovery was reduced by the cryogel thickness (**Fig. 1c of the original manuscript**). However, the structural damage on the device surface was reduced with the thicker overlayer. Accordingly, the J_{ph} degradation rate during the long-term operation, obtained by fitting the time- J_{ph} curve from 1h to the final time points of each device, was lower with the thicker cryogel overlayer (**Supplementary Fig. 6 in the original manuscript**).

Assuming that the whole current of the device involved the Faraday reaction, the overall gas production efficiency calculated by the area of the time-current density graph for the entire duration in **Figs. 1c-d** was similar for CG-800 μm and CG-1200 μm (**Fig. R8**).

Fig. R8. Area of the time-current density graph of *CG-400µm*, *CG-800µm*, and *CG-1200µm* for the duration of 0~5 h, 0~50 h, 0~130 h, 0~170 h, and 0~200 h. Assuming that 100% of the current in the device involves the Faraday reaction, the area can indicate the relative total gas production efficiency.

In the revised manuscript, we modified the confusing statements regarding the gas transport in the cryogel overlayer. We included the statement comparing the gas production performance of the devices with varied cryogel thicknesses. **Fig. R8** was included as **Supplementary Fig. 5** in the revised manuscript. We also modified confusing statements regarding the gas transport in the cryogel overlayer.

Revision made (colored in blue):

(in Page 7)

~~In addition, a thicker overlayer was favorable for extending the lifetime of the PEC device. Compared to *CG-800µm*, the current degradation occurred over the entire duration was reduced in *CG-1200µm*. However, both devices exhibited a similar total gas production, which was higher than that of *CG-400µm* (Supplementary Fig. 5). On the other hand, the initial J_{ph} drop recovery was not observed for *CG-100µm* and *CG-200µm*, which exhibited relatively poor device lifetimes (Supplementary Fig. 6a).~~

(in Page 7)

In our previous study, we reported the 100-h stability ($J_{ph}/J_0 \sim 70\%$) of a Pt/TiO₂/Sb₂Se₃ photocathode using a nanoporous polyacrylamide hydrogel protector that does not possess interconnected micropores for bubble escape²⁵. An increase in the thickness of the hydrogel protector induced severe bubble accumulation, which increased the J_{ph} degradation rate during long-term operation. ~~Despite the microporous cryogelated overlayer helped gas transport, bubbles remained trapped in the overlayer could lead to reduction of current density. In contrast, instead of accumulating in the overlayer, bubbles in the cryogelated overlayer with interconnected micropores could effectively escape, even when the thickness was significantly increased.~~

Comment 11:

Since the reaction temperature is generally above room temperature, does the irradiated light influence the cryogelated overlayer, such as melting or enlarging the pore size.

Response:

We appreciate the reviewer's comment regarding the effect of temperature and irradiated light on the structure of the cryogel. According to the previous study (*Polymer* **44**, 1331-1337 (2003)), the exposure of polyacrylamide hydrogel to hot water with a temperature of 90°C also did not degrade the polymeric network. In addition, previous studies (*Polymer* **44**, 1331-1337 (2003); *Journal of Chromatographic Science* **37**, 486-494 (1999)) showed that the polyacrylamide hydrogel was not significantly degraded under the illumination of UV light with the wavelength of 254 nm and sunlight for more than ten days. Also, we observed that the polyacrylamide hydrogel was structurally stable inside the 0.1M H₂SO₄ electrolyte under light illumination for 1000 h (**Fig. R4**). For this reason, we expected that the irradiated light during the PEC operation might not significantly change the porous structure of the cryogel.

Fig. R4. Optical microscopic images of the polyacrylamide hydrogel after immersion in a 0.1 M H₂SO₄ solution for 0, 24, and 1000 h. Red dotted lines represent the boundaries of the hydrogel.

In the revised manuscript, we have included the statement regarding the stability of polyacrylamide hydrogel under light illumination.

Revision made (colored in blue):

(in Page 6)

The polymeric part of the cryogel overlayer is polyacrylamide hydrogel which is nanoporous, poroelastic, water-permeable, hydrophilic, and aerophobic^{25,29,30}. The polyacrylamide hydrogel was tested to be stable under light illumination^{31,32}.

Comment 12:

From Supplementary Fig. 7, the pore size on the surface of PEC devices has a large correlation with the ion transport and the contact area of the electrolyte. How the experiment ensures a consistent distribution of contact area between the different PEC devices and the cryogelated overlayers.

Response:

We appreciate the reviewer's comment regarding the consistency of the contact area. The polymeric component of the cryogel covers the surface of the PEC devices, and thus the electrolyte can contact the device through the micropores of the cryogel near the device surface. Therefore, we agree with the reviewer that the pore size can be critical in determining the contact area of the electrolyte.

The pore structure of the cryogel depends on the temperature and its gradient of the pre-gel solution (*Chemical Engineering Science* **61**, 6701-6708 (2006), *Advanced Materials* **33**, 2100091 (2021)). To be consistent, we prepared the sample in the same location with a constant temperature and minimal temperature gradient in the freezer. The average area fraction of the electrolyte transportable micropores was estimated to be approximately 59.3 ± 3.5 % for cryogels (**Fig. R9**).

Fig. R9. Area fraction of micropores of the cryogel overlayer obtained from the image-based analysis of the sectional image of the cryogel. The error bars represent one standard deviation (number of samples = 3).

We also note that the similar values in the initial photocurrent density for devices coated by cryogel with varying thicknesses can be indicative of the consistent contact area of the electrolyte between the different PEC devices and the cryogelated overlayers (**Supplementary Fig. 4 of the original manuscript**).

In addition, the attachment of the cryogelated overlayer and PEC device was conducted consistently using APTES and glutaraldehyde as described in Method.

In the revised manuscript, we have included the statement regarding the consistent area of pores in the cryogel and experimental methods to ensure the consistency of the porous structure of the cryogel. **Fig. R9** was included as **Supplementary Fig. 3** in the revised manuscript.

Revision made (colored in blue):

(in Page 6)

The average area fraction of the micropores was estimated to be approximately 59.3 ± 3.5 % for cryogels (**Supplementary Fig. 3**). The polymeric part of the cryogel overlayer is polyacrylamide hydrogel which is nanoporous, poroelastic, water-permeable, hydrophilic, and aerophobic^{25,29,30}.

(in Page 19)

1 ml of pre-gel solution was prepared by mixing 888 μ l of DI water, 65.2 μ l of 100% AAM, 40.8 μ l of 2% BAAM, 5 μ l of 10% AP, and 1 μ l of TEMED. **The cryogelation process was conducted at the same location with a minimal temperature gradient in a freezer.** After cryogelation in a – 20 °C freezer in the laboratory for 12, 24, 48, or 72 h, the rubber mold and cover glass were carefully removed from the cryogelated overlayer.

Reviewers' comments:

Reviewer #2 (Remarks to the Author):

In the revised MS, the authors now emphasize that they investigate the protection mechanism by cryogelated overlayers, and extend PEC device operation by optimizing the overlayer structure. From this perspective, the study could provide sufficient novelty and impact to be considered for Nature Communications. However, a major concern is that the degradation mechanism of the studied system Pt/TiO₂/Sb₂Se₃ is not as the authors portray (i.e. not purely stress-initiated). Specific points are provided below:

1) The authors suggest that the bubbles induce shear stress on the electrode surface, which leads to detachment and agglomeration of the Pt catalyst, followed by reductive dissolution of the TiO₂ layer, and consequently to electrode degradation. However, Figure S7d shows no sign of Pt aggregation, but electrode dissolution still occurs. Similarly, Figures 1f (bottom) and 4b show no sign of Pt aggregation, in contrast to Figure 5a (SEM).

2) It is not clear how the H₂ bubble-induced shear stress could cause Pt aggregation at such a massive scale, as shown in Figure 5a (but not in other pictures). Assuming that the bubbles are produced locally/periodically on the surface, it is unreasonable to suggest that the pressure/stress also increases locally in a non-confined space – such pressure gradient would be expected to nearly instantly dissipate in the photoreactor volume (as it naturally happens during explosions). Therefore, the authors should provide a physical mechanism (or local pressure measurements) that explains how exactly a slow formation (i.e. far from explosive) of gas bubbles can create local stress. The role of water surface tension at the water-bubble interface could be also important.

3) It is not mentioned if the authors used air-saturated or de-aerated solutions in the PEC tests. If oxygen was present in the system, it could be the factor in electrode degradation (via the production of reactive oxygen species or H₂O₂).

4) Are Pt layer aggregation/detachment and TiO₂ dissolution observed during electrochemical water splitting under strongly cathodic conditions (in dark), or under open-circuit photocatalysis conditions? The authors should clarify this experimentally and in the literature.

5) What is the advantage of using such thick (apparently ~10 nm) and continuous Pt layers? How does it affect the optical properties?

6) The depiction of Pt in Figure 1a (as local sites) and Figure 1g (as continuous layer) is confusing – the real surface coverage by Pt should be therefore clarified.

7) Despite the intended applications in solar water splitting, no actual analysis of the evolved gases (e.g. by GC) and of the photonic/Faradaic efficiency of this process is provided.

Reviewer #4 (Remarks to the Author):

I appreciate the authors for their comprehensive responses and the clarifications provided in both the main manuscript and the supplementary materials. They have addressed the comments effectively. However, there are still a few points have to discuss before the manuscript can be accepted.

The following issues require further modification or clarification.

1. The response to Comment 2 from Reviewer 3: The authors sought to explore the mechanical interaction between the bubble and the cryogelated overlayer during the initial J_{ph} drop recovery and the following J_{ph} fluctuation. To do this, they provided in-situ visualizations of bubbles within the cryogelated overlayer for CG-800 μm . However, the images supplemented in the response to Comment 2 are CG-400 μm . It would be beneficial to supplement the corresponding images for CG-800 μm . Furthermore, for clarity, it would be helpful to label the in-situ bubble images with timestamps corresponding to Fig. 1c.

2. During the early stages of the PEC operation, the authors observed the formation of multiple small bubbles within the PEC device. This was attributed to the heterogeneity in the shape and dimension of the pores. As the operation progressed, these small bubbles coalesced to form a larger bubble that remained on the electrode surface. This behavior was evident in both CG-400 μm (Supplementary Fig. 10) and CG-800 μm (Fig. 2a). Interestingly, for CG-1200 μm , the later stages of the PEC operation showed a smaller bubble compared to those in CG-400 μm and CG-800 μm , alongside a cluster of small bubbles (Supplementary Fig. 6b). What could account for this variation? Could it be linked to the material's wettability, as discussed in Joule 5, 887-900 (2021)?

3. The manuscript's analysis indicates that a larger thickness of the cryogelated overlayer corresponds to a greater degree of pore coalescence, denoted as λ_{pore} . An increase in λ_{pore} leads to an increase in the reduction in V_{bubble} at the initial escape, while simultaneously causing a decrease in the fluctuation amplitude of V_{bubble} (Fig. 3d, Fig. 3e). When examining the relationship with the current, a larger λ_{pore} suggests that the associated current density drop recovery will be significant if a pore fractures. Subsequently, the periodic fluctuation of the current density will diminish after a pore fractures. Contrarily, the experimental data for GC-1200 μm in the manuscript shows a relatively minor current density drop recovery (Fig.1c). This observation appears to contradict the manuscript's conclusions. Could you provide an explanation for this discrepancy?

Some statements need to be checked and modified.

1. Comment on (previously) Page 23: I believe the authors have an error in their description of the degree of J_{ph} drop. Instead of stating, "The degree of J_{ph} drop was determined as the photocurrent density at the time just before the occurrence of the photocurrent recovery divided by the initial photocurrent density," it should be revised to: "The degree of J_{ph} drop is calculated as the difference between the initial photocurrent density and the photocurrent density right before the occurrence of the photocurrent recovery, all divided by the initial photocurrent density."

2. Manuscript (Page 9, Line 2): The assertion, "We observed that the amplitude of the J_{ph} fluctuation was small when the initial J_{ph} drop recovery was large, and the J_{ph} degradation during the long-term operation was faster when the J_{ph} fluctuation was large," seems to contradict the experimental findings

presented. Specifically, for CG-1200 μm , both the initial Jph drop recovery and the Jph fluctuations are minimal (Fig. 1c).

3. Manuscript (Page 2, Line 9): The statement, “Tests for cryogel overlayers with varied microstructures reveal that the hydrogen gas trapped in the cryogel protector reduce shear stress at the catalyst surface by providing bubble nucleation sites.” can be misinterpreted. The phrase “cryogel overlayers with varied microstructures” might be misconstrued as “cryogel overlayers with different pore structures.” However, throughout this paper, the pore structure of the cryogel overlayers remain consistent.

Point-by-point response to reviewers' comments

Title: Stable water splitting using photoelectrodes with a cryogelated overlayer

Author(s): Byungjun Kang[†], Jeiwan Tan[†], Kyungmin Kim, Donyoung Kang, Hyungsoo Lee, Sunihl Ma, Young Sun Park, Juwon Yun, Soobin Lee, Chan Uk Lee, Gyumin Jang, Jeongyoub Lee, Jooho Moon*, Hyungsuk Lee*

<Reviewer #2>

Remark:

In the revised MS, the authors now emphasize that they investigate the protection mechanism by cryogelated overlayers, and extend PEC device operation by optimizing the overlayer structure. From this perspective, the study could provide sufficient novelty and impact to be considered for Nature Communications. However, a major concern is that the degradation mechanism of the studied system Pt/TiO₂/Sb₂Se₃ is not as the authors portray (i.e. not purely stress-initiated). Specific points are provided below:

Response:

We express our sincere gratitude to the reviewer for the review and evaluation of our work. As the reviewer commented, we believe that investigation of the protection mechanism of the device-on-top porous overlayer protector and extension of the durability of the PEC device by the optimized cryogel could provide sufficient novelty and impact to the PEC research field. Also, we firmly believe that the reviewer's comments helped us to improve the quality of our manuscript.

We agree with the reviewer's major concern that the degradation of the studied system Pt/TiO₂/Sb₂Se₃ is 'not purely stress initiated.' Typically, the lifetime of PEC devices is hampered by 1) severe photocorrosion of semiconductors and protection layers and 2) instability of co-catalysts. These two types of structural degradation occur simultaneously in a PEC system, such as photoelectrode. For example, in our previous study, we observed both of the Pt layer

aggregation/detachment and the partial dissolution of the TiO₂ layer in the Pt/TiO₂/Sb₂Se₃ device when the J_{ph}/J_0 reached $\sim 70\%$ (*Nat Energy* 7, 537–547 (2022)). Although we modified our manuscript during the first revision, there are still a few confusing statements from which the reviewer’s comment might have arisen. In the revised manuscript, we modified the unclear statements regarding the degradation mechanism of the studied system Pt/TiO₂/Sb₂Se₃.

In addition, during this revision, we also found that while the structure of the Pt layer was not severely damaged (**Fig. 1g**), the thickness of the TiO₂ layer of the *CG-1200 μ m* could be decreased by approximately 8% due to the reductive dissolution (**Fig. R1**).

Fig. R1. The thickness of the TiO₂ layer in the as-prepared Pt/TiO₂/Sb₂Se₃ device with a J_{ph}/J_0 of 100% and the *CG-1200 μ m* at a J_{ph}/J_0 of 30%. The thickness of TiO₂ layer was manually characterized from the STEM-EDS element mapping data for Ti.

In the revised manuscript, we included **Fig. R1** as **Supplementary Fig. 10** and mentioned the decrease of the TiO₂ layer thickness in the *CG-1200 μ m* at a J_{ph}/J_0 of 30%.

Revision made (colored in blue):

(in Page 3)

Despite uniform TiO₂ deposition on semiconductor, agglomeration and detachment of the co-catalyst from the photoelectrodes can accelerate **the reductive dissolution and** the photocurrent degradation reducing the ~~device~~-lifetime **of device**.

(in Page 8)

This might be attributed to the detachment of the Pt catalyst from the device ~~followed by with~~ the chemical dissolution of TiO₂/Sb₂Se₃ nanorods^{10,25} (**Supplementary Fig. 7d Figs. 9d and 9e**). ... This result suggested that the direct spatial confinement of catalyst nanoparticles by the nanoporous overlayer²⁵ might not play a critical role in stabilizing the catalysts. **The TiO₂ layer of the CG-1200 μ m experienced an approximate 8% decrease in thickness (Supplementary Fig. 10) due to the reductive dissolution.** ~~In addition,~~ The crystallinity and orientation of Sb₂Se₃ were maintained, and no secondary phases were observed even after the PEC operation over 200 h (**Supplementary Fig. 811**).

(in Page 13)

The shear stress on the catalyst on the device surface could lead to detachment and agglomeration of the Pt catalyst¹⁶ and **subsequent** reductive dissolution of the TiO₂ layer²⁹.

(in Page 14)

This might be due to the detachment and agglomeration of the Pt catalyst¹⁰, ~~followed by and the~~ reductive dissolution of the TiO₂ layer. ... Suppression of bubble nucleation at the device surface might contribute to preserving the Pt co-catalyst surface ~~by preventing shear stress induced structure degradation~~.

The point-by-point responses to the reviewer's comments can be found below. The 'original manuscript' and 'revised manuscript' in our responses refer to the first revised version of the manuscript and the one revised again based on it, respectively.

Comment 1:

The authors suggest that the bubbles induce shear stress on the electrode surface, which leads to detachment and agglomeration of the Pt catalyst, followed by reductive dissolution of the TiO₂ layer, and consequently to electrode degradation. However, Figure S7d shows no sign of Pt aggregation, but electrode dissolution still occurs. Similarly, Figures 1f (bottom) and 4b show no sign of Pt aggregation, in contrast to Figure 5a (SEM).

Response:

We appreciate the reviewer's comment regarding the electron microscopy images of the device surface during the degradation. The SEM images mentioned by the reviewer were captured at a low magnification to observe the morphological changes in multiple rod-like structures of the device over a region having an area of several square micrometers. It is technically difficult to clarify the small nanoparticles, such as the Pt co-catalyst nanoparticles having a size under 10 nm, in these low-magnification images. The 'no sign of Pt aggregation' in the SEM images for the device that experienced the reductive dissolution of TiO₂ (**Supplementary Fig. 7 and Fig. 4b of the original manuscript**) could suggest either that the sizes of the aggregated Pt particles were not large enough to be visible in the low-magnification SEM images or that the Pt particles had become detached from the device.

The cross-sectional high-resolution transmission electron microscopy (TEM) of the device clearly demonstrated that the 3-5 nm sized Pt nanoparticles were agglomerated to 10-20 nm sized particles or detached from the device when the photocurrent (J_{ph}) of the device was reduced to 70% of the initial photocurrent (J_0) (**Fig. R2a**). The 10-20 nm sized agglomerated Pt particles did not exhibit a distinct particle morphology in the SEM image (**Fig. R2b**).

The structural damage to these Pt co-catalysts occurs concurrently with the chemical dissolution of the TiO₂ layer which was located under the Pt particles (*Nat Energy* 7, 537–547 (2022)). After

the significant dissolution of the TiO₂ layer, the Pt particles can be detached from the whole device and thus not shown in the SEM images (right image of **Fig. 4b** and fourth image of **Supplementary Fig. 7** of the original manuscript). When the device surface was not severely dissolved and the Pt detachment was suppressed by the overlayer, the Pt particles could be agglomerated to 50-100 nm sized large particles that are visible in the SEM images (**Fig. 5a of the original manuscript**). The size of the agglomerated Pt nanoparticles can also be regulated by the mechanical stress applied on the device surface.

The aggregation and detachment of Pt co-catalyst would not be shown in the SEM images for the devices without significant degradation in the surface structure of the device, as the *CG-1200 μ m* (**bottom images of Fig. 1f**) or center region of the *CG-400 μ m* (**left image of Fig. 4b**).

Fig. R2. The change of the structure of the Sb_2Se_3 device during the degradation in photocurrent. (a) Cross-sectional transmission electron microscopy (TEM) images and (b) surface scanning electron microscopy (SEM) images of the as-prepared ($J_{ph}/J_0 \sim 100\%$) and degraded ($J_{ph}/J_0 \sim 70\%$) Sb_2Se_3 device. The scale bars in the TEM and SEM images are 10 nm and 500 nm, respectively. The data in **Fig. R2b** were obtained from **Supplementary Fig. 7d** in the original manuscript.

In the revised manuscript, we stated that the Pt catalyst could be severely agglomerated into large particles while being covered with the cryogel and thus could be observed as bright spots via the SEM image in **Fig. 5a**. To clearly demonstrate the structural damage of the Pt catalyst, we have included the left and right images in **Fig. R2a** in the inset of **Fig. 1f** and **Supplementary Fig. 9e** of the revised manuscript, respectively. The zoomed-in images in **Fig. R2b** were included in **Supplementary Fig. 9d** of the revised manuscript.

Revision made (colored in blue):

(in Page 14)

However, after the operation, some bright spots were observed in the scanning electron microscopy (SEM) images of the device surface, ~~indicating agglomeration of the Pt catalyst~~. This result indicated that the Pt particles were continuously aggregated to form large particles while the Pt detachment was suppressed by the overlayer.

(Fig. 1)

Fig. 1. Cryogelated polyacrylamide overlayer for photoelectrochemical (PEC) devices. ... (f) The surface microstructure of an as-fabricated reference PEC device and CG-1200 μm at a J_{ph}/J_0 of 30%, where J_0 represents the initial photocurrent density. The scale bar represents 500 nm. ~~The inset image in the top image is the cross-sectional transmission electron microscopy (TEM) image of the reference PEC device. The scale bar in the inset image represents 10 nm.~~

(Supplementary Fig. 9)

Supplementary Fig. 9. Preservation of the photoelectrochemical performance and surface morphology of the photoelectrochemical (PEC) device by the cryogelated overlayer. ... (d) Scanning electron microscopy (SEM) images of the device surface of *no cryogel* when J_{ph}/J_0 was 100%, 70%, 50%, and 20%. **(e) Cross-sectional transmission electron microscopy image of *no cryogel* when J_{ph}/J_0 was 70%.** SEM images of (f) *CG-400 μm* and (g) *CG-800 μm* captured after peeling off the overlayer.

Comment 2:

It is not clear how the H₂ bubble-induced shear stress could cause Pt aggregation at such a massive scale, as shown in Figure 5a (but not in other pictures). Assuming that the bubbles are produced locally/periodically on the surface, it is unreasonable to suggest that the pressure/stress also increases locally in a non-confined space – such pressure gradient would be expected to nearly instantly dissipate in the photoreactor volume (as it naturally happens during explosions). Therefore, the authors should provide a physical mechanism (or local pressure measurements) that explains how exactly a slow formation (i.e. far from explosive) of gas bubbles can create local stress. The role of water surface tension at the water-bubble interface could be also important.

Response:

We appreciate the reviewer's comments regarding the generation of mechanical stress by the bubble. We note that we provided evidence of the Pt aggregation on the device surface in our response to your comment 1.

When the bubbles are generated by the electrochemical devices, they are grown on the device surface. The bubbles can be detached from the device when their size reaches a critical value (*J Electrochem Soc* **164**, E448-E459 (2017)). As shown in **Fig. R3**, the bubbles attached on the device surface can exert mechanical forces on the device surface (*Int J Heat Mass Transfer* **54**, 3234-3244 (2011); *J Electrochem Soc* **164**, E448-E459 (2017); *Colloids Surf A Physicochem Eng Asp* **653**, 130008 (2022)). Those forces by the bubbles can be regulated by the surface energy at the interfaces of bubbles, solution, and PEC device (*Soft Matter* **5**, 3963-3968 (2009); *J Electrochem Soc* **164**, E448-E459 (2017)), as the reviewer commented. When the bubbles are detached from the device, they would no longer apply mechanical stress on the surface of the device unless they explode, as the reviewer mentioned.

Fig. R3. Schematic of the physical forces applied on the bubbles and the electrochemical or photoelectrochemical devices.

Bubbles can be nucleated at various regions of the device, expand from a few nanometers to a size over tens of micrometers, and travel distances of over hundreds of micrometers before being detached from the device (*Nat Energy* 7, 537–547 (2022)). Thus, the physical forces by a number of hydrogen bubbles repeatedly generated during hours of the PEC operation can cause structural damage to the Pt co-catalyst over a wide region of the device surface.

In addition, for the cryogel-protected PEC devices, nucleated bubbles were trapped in the cryogel and thus highly confined by the porous polymeric structure of the cryogel. When the trapped bubble is expanded, the bubble can apply pressure and mechanical stress to the surrounding material (*Soft Matter* 5, 3963-3968 (2009)), such as the overlayer and PEC device (**Figs. R4a and R4b**). As the distance between the bubble and the device decreases, the magnitude of the mechanical stress exerted on the device by the bubble increases (**Fig. R4c**). Thus, the bubbles trapped near the device surface (**Fig. 5a**) have a potential to exert mechanical stress on the catalyst layer, leading to the substantial structural damage of the catalyst.

Fig. R4. Numerical analysis of the mechanical stress applied on the PEC device by the expansion of bubble inside the cryogel overlayer. (a) Schematic of the numerical simulation. (b) Finite-element method (FEM) results of the mechanical stress applied on the overlayer and PEC device during the expansion of bubble confined in the overlayer. (c) FEM result of the mechanical stress on the overlayer and PEC device with varying distance between the bubble and PEC device. The scale bars represent 50 μm . The elastic modulus of the PEC device and overlayer were assumed to be 50 GPa and 10 kPa, respectively.

In the revised manuscript, we included the statement regarding the generation of mechanical stress by the bubbles on the PEC device and confined in the overlayer. The **Figs. R3 and R4** were included as **Supplementary Figs. 1 and 20**, respectively, in the revised manuscript.

Revision made (colored in blue):

(in Page 4)

The surface-growing bubbles apply mechanical force to the device surface **at the bubble-device interface**¹⁶ (**Supplementary Fig. 1**), which may cause migration and aggregation of the co-catalysts^{17,18}.

(in Page 13)

We speculated that the mechanical interaction with the cryogel occurred during bubble dynamics and consequential shear stress on the device surface would be less for microbubbles **confined at a distance from the device surface** due to their smaller dimension.

(in Page 14)

It is speculated that the **initial bubble expansion of bubbles** nucleated in the isolated micropores near the device surface, ~~followed by the translation of additional hydrogen gas, which contributed to the bubble growth producing the shear stress to the Pt catalyst~~¹⁷ can generate mechanical stress to the PEC device (**Supplementary Fig. 20**).

Comment 3:

It is not mentioned if the authors used air-saturated or de-aerated solutions in the PEC tests. If oxygen was present in the system, it could be the factor in electrode degradation (via the production of reactive oxygen species or H₂O₂).

Response:

We appreciate the reviewer's important comment to clarify our work. In all PEC characterizations, we purged our electrolytes with inert Ar gas to prevent the oxygen reduction reaction. In the revised manuscript, we included the statement regarding the purging of electrolytes with Ar gas.

Revision made (colored in blue):

(in Page 20, Methods Section)

The three electrodes were submerged in **a an Ar-purged** 0.1 M H₂SO₄ electrolyte (pH = 1),

Comment 4:

Are Pt layer aggregation/detachment and TiO₂ dissolution observed during electrochemical water splitting under strongly cathodic conditions (in dark), or under open-circuit photocatalysis conditions? The authors should clarify this experimentally and in the literature.

Response:

We appreciate the reviewer's thoughtful comment. Applying a strongly cathodic condition in the dark (*i.e.*, highly negative potential) would result in a high dark current flow accompanying a large amount of hydrogen bubble production. Aggregation and detachment of Pt catalysts were frequently observed regardless of support materials (*ACS Catal* **12**, 20, 13021–13033 (2022); *J Mater Chem A*, **8**, 16582-16589 (2020); *ACS Catal* **2**, 5, 832–843 (2012)). In addition, according to the Pourbaix diagram of TiO₂, strongly cathodic conditions, such as negative potential, will cause the reduction of the TiO₂ layer (*J Electrochem Soc* **160**, C277–C284 (2013)). However, since the strongly cathodic condition in the dark is not a practical and conventional operation condition for photoelectrodes, we did not characterize the PEC devices in this condition in detail.

Instead, we conducted additional dark electrolysis using the Pt/FTO electrodes and observed microstructures of the Pt layer before/after electrolysis. The deposition conditions for the Pt layer on the Pt/FTO electrode were the same as those used to fabricate the Pt/TiO₂/Sb₂Se₃ device. The top-view SEM images of FTO and Pt/FTO (as-prepared, before electrolysis) were almost identical (**Figs. R5a and 5b**), indicating that the Pt nanoparticles were not visible in the SEM image due to their small size. The uniformly deposited Pt nanoparticles with a size of 3-5 nm were found in the TEM image of the cross-sectioned sample of the Pt/FTO electrode (**Fig. R5c**).

During the dark electrolysis of the Pt/FTO electrode at -0.1 V_{RHE}, the current of the electrode was significantly degraded within 2 h (**Fig. R5d**). Since there is no possibility of the photocorrosion of semiconductors for the Pt/FTO electrode, the current degradation of the electrode could be originated from the physical damage of the Pt layer. In the SEM image captured after 2 h of operation, we observed partial structural damage of the surface of the Pt/FTO device, and the aggregation of Pt particles in some regions (**Fig. R5e**).

Fig. R5. Characterization of the structural damage of Pt catalyst under the strongly cathodic condition. The top-view SEM images of (a) FTO and (b) Pt/FTO. Scale bars represent 100 nm. (c) The TEM image of cross-sectioned Pt/FTO. Scale bar represents 20 nm. (d) Time-current density profile during the chronoamperometry measurement of Pt/FTO. (e) The top-view SEM images of Pt/FTO after 2 h of operation. Scale bar represents 200 nm.

The current obtained under the light illumination in the photocathodes can be considered as a “photo-assisted” current at the standard potential of hydrogen production. Thus, most of the studies on photocathodes for water-splitting characterize the device stability at 0 V_{RHE} rather than the open circuit condition (*Chem Soc Rev* **46**, 1933-1954 (2017)), as in Fig. 1c in our manuscript.

In the revised manuscript, we included the results of the experimental analysis of the structural damage of the Pt catalyst under the strongly cathodic condition as **Supplementary Note 4**, and **Fig. R5** was inserted as **Supplementary Note Fig. 6**.

Revision made (colored in blue):

(in Page 18, Methods Section)

Finally, the Pt co-catalyst was sputtered using a sputter coater (Ted Pella, Redding, CA, USA) at an applied current of 10 mA for 120 s. The Pt/FTO electrodes were also fabricated using the same deposition procedure to characterize the optical, structural, and electrochemical properties of Pt co-catalyst layer prior to coating onto the photoelectrodes. (**Supplementary Note 4**).

(Supplementary Note 4)

Supplementary Note 4. Characterization of the optical, structural, and electrochemical properties of Pt catalyst.

We fabricated the Pt/FTO electrode using the Pt deposition condition which was the same to those used in the fabrication of the Pt/TiO₂/Sb₂Se₃ PEC device. The UV-vis transmittance spectrum of the Pt/FTO electrode normalized by a bare FTO substrate exhibited the optical transmittance of 90% in average (**Supplementary Note Fig. 5**). The top-view SEM images of FTO and Pt/FTO (as-prepared, before electrolysis) were almost identical (**Supplementary Note Figs. 6a and 6b**), indicating that the Pt nanoparticles were not visible in the SEM image due to its small size. The uniformly deposited Pt nanoparticles with a size of 3-5 nm were found in the TEM image of the cross-sectioned sample of the Pt/FTO electrode (**Supplementary Note Fig. 6c**).

During the dark electrolysis of Pt/FTO electrode at -0.1 V_{RHE}, the current of the electrode was significantly degraded within 2 h (**Supplementary Note Fig. 6d**). Since there is no possibility of the photocorrosion of semiconductors for the Pt/FTO electrode, the current degradation of the electrode could be originated from the physical damage of Pt layer. After 2 h of the operation, we observed partial structural damage of the surface of the Pt/FTO device, and the aggregation of Pt particles in some regions (**Supplementary Note Fig. 6e**).

Supplementary Note Fig. 6. Characterization of the structural damage of Pt catalyst under the strongly cathodic condition. The top-view SEM images of (a) FTO and (b) Pt/FTO. Scale bars represent 100 nm. (c) The TEM image of cross-sectioned Pt/FTO. Scale bar represents 20 nm. (d) Time-current density profile during the chronoamperometry measurement of Pt/FTO. (e) The top-view SEM images of Pt/FTO after 2 h of operation. Scale bar represents 200 nm.

Comment 5:

What is the advantage of using such thick (apparently ~10 nm) and continuous Pt layers? How does it affect the optical properties?

Response:

We appreciate the reviewer's comment on the structure and optical properties of the Pt layer on the PEC device. We expect that this comment might be attributed to our unclear description of the particulate structure of the Pt layer in our original manuscript.

As mentioned in our response to your comment 1, the Pt layer on the PEC device is consisted of 3-5 nm-sized particles, not a continuous film (**Fig. R2a**). The structure of the Pt layer was also clearly demonstrated in the cross-sectional TEM image of the Pt/FTO device prepared with the same Pt deposition condition (**Fig. R6a**). The Pt layer of the Pt/FTO device showed ~ 90% transmittance on average (**Fig. R6b**), indicating that the Pt co-catalyst is suitable as a top co-catalyst layer of photocathodes.

As reported in previous literatures, a thicker Pt layer is beneficial for the stability of photoelectrodes (*Adv Energy Mater* **8**, 1702888 (2018)). However, the thicker layer of the Pt co-catalyst decreased the amount of light transmitted into the absorbing layer, *i.e.*, semiconductor, and reduced the photocurrent densities of photoelectrodes (*Sustain Energy Fuels* **3**, 2227-2236 (2019)).

Fig. R6. Characterization of the structure and optical properties of the Pt co-catalyst layer. (a) Cross-sectional transmission electron microscopy (TEM) image of the Pt/FTO device. The scale bar represents 20 nm. (b) UV-vis transmittance spectrum of the Pt/FTO electrode. The bare FTO substrate was utilized as a reference sample. The Pt layer deposition conditions onto the Pt/FTO electrode were consistent with those employed during the fabrication of the Pt/TiO₂/Sb₂Se₃ PEC device.

When the Pt nanoparticles are overlapped or slightly agglomerated, the Pt nanoparticles can be seen as a continuous film-like structure as in the **inset image of Fig. 1g of the original manuscript (left zoomed-in image of Fig. R7)**. In the other region of the image in Fig. 1g, we could observe the particulate structure of the Pt co-catalyst (**right zoomed-in image of Fig. R7**).

Fig. R7. Cross-sectional TEM image of the CG-1200 μ m at a J_{ph}/J_0 of 30%. The scale bar represents 100 nm. The scale bars in the zoomed-in images represent 20 nm. The data was from Fig. 1g of the original manuscript.

In the revised manuscript, we included the statement regarding the optical properties of the particulate layer of Pt co-catalyst. **Fig. R6a** was included as **Supplementary Note Fig. 6c**. The data in **Fig. R6b** was included as **Supplementary Note Fig. 5**. To clearly demonstrate the particulate structure of the Pt co-catalyst, we added the cross-sectional TEM image of the reference device in **Fig. R2a** as an inset image in **Fig. 1f**. The inset image of **Fig. 1g** was replaced by the right image of **Fig. R7**.

Revision made (colored in blue):

(Supplementary Note 4)

We fabricated the Pt/FTO electrode using the Pt deposition condition which was the same to those used in the fabrication of the Pt/TiO₂/Sb₂Se₃ PEC device. The UV-vis transmittance spectrum of the Pt/FTO electrode normalized by a bare FTO substrate exhibited the optical transmittance of 90% in average (**Supplementary Note Fig. 5**).

(Supplementary Note Figure 5)

Supplementary Note Fig. 5. UV-vis transmittance spectrum of the Pt/FTO electrode. The bare FTO substrate was utilized as a reference sample.

(Figure 1)

Fig. 1. Cryogelated polyacrylamide overlayer for photoelectrochemical (PEC) devices. ... (f) The surface microstructure of an as-fabricated reference PEC device and $CG-1200\mu\text{m}$ at a J_{ph}/J_0 of 30%, where J_0 represents the initial photocurrent density. The scale bar represents 500 nm. **The inset image in the top image is the cross-sectional transmission electron microscopy (TEM) image of the reference PEC device. The scale bar in the inset image represents 10 nm. (g) Cross-sectional TEM image...**

Comment 6:

The depiction of Pt in Figure 1a (as local sites) and Figure 1g (as continuous layer) is confusing – the real surface coverage by Pt should be therefore clarified.

Response:

We appreciate the reviewer’s comment regarding the description of the surface coverage by Pt in the schematic of **Fig. 1a of the original manuscript**. The cross-sectional TEM images of the device (**Fig. R2a**) suggested that the Pt co-catalyst was deposited on the PEC device as nanoparticles rather than a continuous layer, as depicted in **Fig. 1a of the original manuscript**.

As per the reviewer’s suggestion, we modified the schematic to depict the surface coverage by Pt more clearly in **Fig. 1a of the revised manuscript**. We also clarified that the schematic was not drawn to scale.

Revision made (colored in blue):

(Figure 1)

Fig. 1. Cryogelated polyacrylamide overlayer for photoelectrochemical (PEC) devices. (a) Schematic illustration of the three-dimensional and cross-sectional structure of the device with a cryogelated overlayer. **The schematic was drawn not to scale. ...**

Comment 7:

Despite the intended applications in solar water splitting, no actual analysis of the evolved gases (e.g. by GC) and of the photonic/Faradaic efficiency of this process is provided.

Response:

We appreciate the reviewer's comment to improve the quality of our manuscript. As per the reviewer's comment, we additionally performed the gas chromatography (GC) and incident photon-to-current conversion efficiency (IPCE) measurements.

The amount of photoelectrochemically produced hydrogen gas was measured during the operation of *CG-400 μm* and *CG-1200 μm* by the gas chromatography analysis. The Faradaic efficiency was close to 100 % except during the initial J_{ph} drop within 30 minutes (**Fig. R8**).

Fig. R8. Hydrogen production as a function of operation time during the stability test for the *CG-400 μm* and *CG-1200 μm* . The solid line represents the calculated amount of gas produced by the devices, assuming 100% Faradaic efficiency, based on the measured current for each device. The symbols indicate the amount of gas production determined through experimental gas chromatography analysis.

We also characterized the IPCE spectra for the freshly made *no cryogel*, *CG-400 μm* , *CG-800 μm* , and *CG-1200 μm* devices. Compared to the *no cryogel*, three devices with cryogel overlayer exhibited lower IPCE values in the 300–600 nm region, whereas they showed higher values in the 600–1000 nm region. The integrated IPCE value, calculated by integrating the values across all wavelength ranges, did not significantly change by the cryogel coating. The integrated IPCE values for all devices (**Fig. R9**) were similar to the initial current densities of the devices measured via chronoamperometry (**Supplementary Fig. 4 of the original manuscript**).

Fig. R9. The incident photon-to-current conversion efficiency (IPCE) spectra at 0 V_{RHE} for *No Cryogel*, *CG-400 μ m*, *CG-800 μ m*, and *CG-1200 μ m*. The integrated IPCE values for all devices using the solar AM 1.5 spectrum were similar to the initial values obtained prior to J_{ph} drop. The integrated IPCE value, calculated by integrating the values across all wavelength ranges, did not significantly change by the cryogel coating. For *CG-400 μ m*, *CG-800 μ m*, and *CG-1200 μ m*, the IPCE characterization was conducted at the freshly made state before the initial J_{ph} drop occurred.

In the revised manuscript, we have included the IPCE data (**Fig. R9**) and GC data (**Fig. R8**) as **Supplementary Fig. 5b** and **Supplementary Fig. 6**, respectively. We also have added relevant descriptions as follows.

Revision made (colored in blue):

(in Page 7)

After this recovery, the J_{ph} of *CG-400 μ m*, *CG-800 μ m*, and *CG-1200 μ m* was stably maintained for around 30 h, followed by the gradual degradation of J_{ph} (**Fig. 1d**). According to the gas chromatography analysis for the *CG-400 μ m* and *CG-1200 μ m*, the cryogel-coated PEC devices produced the hydrogen gas at a Faradaic efficiency close to 100 % except the initial J_{ph} drop period (**Supplementary Fig. 6**).

(Supplementary Figure 5)

Supplementary Fig. 5. Characterization of the photoelectrochemical characteristics of Sb₂Se₃ photocathodes with and without the cryogel overlayer. (a) The current density-time profile of Sb₂Se₃ photocathodes with and without the cryogelated overlayer measured via chronoamperometry at 0 V_{RHE} under 1-sun illumination during 0 to 0.05 h. (b) The incident photon-to-current conversion efficiency (IPCE) spectra at 0 V_{RHE} for *No Cryogel*, *CG-400μm*, *CG-800μm*, and *CG-1200μm*. The integrated IPCE values for all devices using the solar AM 1.5 spectrum were similar to the initial values obtained prior to J_{ph} drop. The integrated IPCE value, calculated by integrating the values across all wavelength ranges, did not significantly change by the cryogel coating. For *CG-400μm*, *CG-800μm*, and *CG-1200μm*, the IPCE characterization was conducted at the freshly made state before the initial J_{ph} drop occurred.

<Reviewer #4>

Remark:

I appreciate the authors for their comprehensive responses and the clarifications provided in both the main manuscript and the supplementary materials. They have addressed the comments effectively. However, there are still a few points have to discuss before the manuscript can be accepted.

Response:

We would like to express our gratitude to the reviewer for the insightful and critical comments on our manuscript. We believe that the quality of our manuscript was highly improved while addressing the comments raised by the reviewer. The ‘original manuscript’ and ‘revised manuscript’ in our responses refer to the first revised version of the manuscript and the one revised again based on it, respectively. The point-by-point responses to the reviewer’s comments can be found below.

Comment 1 (The following issues require further modification or clarification):

The response to Comment 2 from Reviewer 3: The authors sought to explore the mechanical interaction between the bubble and the cryogelated overlayer during the initial Jph drop recovery and the following Jph fluctuation. To do this, they provided in-situ visualizations of bubbles within the cryogelated overlayer for CG-800 μm . However, the images supplemented in the response to Comment 2 are CG-400 μm . It would be beneficial to supplement the corresponding images for CG-800 μm . Furthermore, for clarity, it would be helpful to label the in-situ bubble images with timestamps corresponding to Fig. 1c.

Response:

We appreciate the reviewer's comment regarding the bubble images in the Figure for the comment 2 of Reviewer 3, which was included as **Supplementary Fig. 10** of the original manuscript during the previous revision, and the label of timestamps corresponding to the chronoamperometry test in **Fig. 1c**.

As the reviewer suggested, instead of the images for the $CG-400\mu m$ in **Supplementary Fig. 10** of the original manuscript, the image for the $CG-800\mu m$ at the earlier timepoint was added as the second image (~ 2 min) in **Fig. 2a** of the revised manuscript. In addition, images in **Fig. 2a** in the revised manuscript now have the label of timestamps.

Revision made (colored in blue):

(Figure 2)

Fig. 2. Effect of cryogelated overlayer on the durability of Sb_2Se_3 photocathodes. (a) Time-lapse images of trapping, expansion, and escape of bubbles in $CG-800\mu m$. White dotted lines in the images represent the boundary of the device surface. **The time at the bubble escape was denoted as 'T'.**

(in Page 9)

At the early stages of the PEC operation, small gas bubbles with different shapes and positions were observed (**Supplementary Fig. 10**) due to the heterogeneity in shape and dimension of pores (**Fig. 1b**).

Comment 2 (The following issues require further modification or clarification):

During the early stages of the PEC operation, the authors observed the formation of multiple small bubbles within the PEC device. This was attributed to the heterogeneity in the shape and dimension of the pores. As the operation progressed, these small bubbles coalesced to form a larger bubble that remained on the electrode surface. This behavior was evident in both CG-400 μm (Supplementary Fig. 10) and CG-800 μm (Fig. 2a). Interestingly, for CG-1200 μm , the later stages of the PEC operation showed a smaller bubble compared to those in CG-400 μm and CG-800 μm , alongside a cluster of small bubbles (Supplementary Fig. 6b). What could account for this variation? Could it be linked to the material's wettability, as discussed in *Joule* 5, 887-900 (2021)?

Response:

We appreciate the reviewer's comment regarding the formation of large bubbles in the cryogel-protected PEC device. First, we note that the data shown in **Supplementary Fig. 6b of the original manuscript** was for the *CG-200 μm* , not the *CG-1200 μm* .

As the reviewer suggested, there was a variation regarding the cluster of small bubbles near the large trapped bubbles as a function of the cryogel thickness. Within the thicker cryogel overlayer, as the distance for large trapped bubbles to travel is increased, the gas contained with the large trapped bubbles in the cryogel not only migrate toward surface pores for escape but also enter adjacent micropores. Since the surface of pore was hydrophilic and aerophobic, gas bubbles were unable to easily pass through the pores and could be trapped to form microbubble clusters shown in **Supplementary Fig. 16c of the original manuscript**. As a result, small bubbles were observed more often at a thicker cryogel overlayer. The dynamic behaviors of bubbles in the porous material, such as coalescence, trapping and escape of bubbles, can be regulated by the wettability of the material as discussed in *Joule* 5, 887-900 (2021).

We agree with the reviewer and have revised the manuscript to state the effect of the material's wettability on the formation of microbubbles.

Revision made (colored in blue):

(in Page 13)

We observed a large number of microbubbles at the boundary of the large bubble in the overlayer for *CG-1200 μ m* during the long-term operation (**Fig. 4c**). The large trapped bubbles in the thick cryogel not only migrate toward surface pores for escape but also enter adjacent micropores. Those gas bubbles were unable to easily pass through the pores and could be trapped to form microbubble clusters due to their aerophobicity.

(in Page 15)

Suppression of bubble nucleation at the device surface might contribute to preserving the Pt co-catalyst surface. The bubble dynamics in the cryogel overlayer including nucleation, coalescence, trapping, and escape of bubbles, which are critical in determining the protective function of the cryogel, can be regulated by the wettability of the porous material³⁸.

Comment 3 (The following issues require further modification or clarification):

The manuscript's analysis indicates that a larger thickness of the cryogelated overlayer corresponds to a greater degree of pore coalescence, denoted as λ_{pore} . An increase in λ_{pore} leads to an increase in the reduction in V_{bubble} at the initial escape, while simultaneously causing a decrease in the fluctuation amplitude of V_{bubble} (Fig. 3d, Fig. 3e). When examining the relationship with the current, a larger λ_{pore} suggests that the associated current density drop recovery will be significant if a pore fractures. Subsequently, the periodic fluctuation of the current density will diminish after a pore fractures. Contrarily, the experimental data for GC-1200 μm in the manuscript shows a relatively minor current density drop recovery (Fig.1c). This observation appears to contradict the manuscript's conclusions. Could you provide an explanation for this discrepancy?

Response:

We appreciate the reviewer's comments on the discrepancy between the theoretical analysis and the experimental observations shown for the *CG-1200 μm* in particular.

Our model was able to predict the reduction in fluctuation amplitude and V_{bubble} change in the initial escape as the thickness of cryogelated overlayer. However, as the reviewer commented, the degree of the J_{ph} recovery of the *CG-1200 μm* was smaller than those of other devices, which was different from the prediction based on the theoretical and numerical analysis. It is because the current density recovery is determined by not only λ_{pore} but also length of gas transport path which becomes more complex as the thickness of overlayer increases.

λ_{pore} represents the diameter of the pore which was fractured near the expanding bubble. However, for escaping of trapped bubbles, they should be transported through not only fractured pore but micropores present in the path for gas transport. We could observe the gas transport through the layer of micropores present between the trapped bubble and the overlayer surface (**Fig. R11a**).

At a given pressure difference in a channel, the gas transport rate is inversely proportional to the path length (L_{path}) that can increased as the overlayer thickness increases according to Poiseuille's Law (*Anaesth Intensive Care Med* **8**, 7-10 (2007)). Additionally, the resistance for gas transport can be increased by the elastic expansion and contraction of the pores. The distensibility of cryogel should be taken into account to estimate an effective value of L_{path} .

Fig. R11. Gas transport through the layer of micropores between the trapped bubble and the overlayer surface. (a) Photographs and schematic of the bubbles in the $CG-1200\mu\text{m}$ during the regular fluctuation of the J_{ph} at ~ 30 h. The scale bar represents 1 mm. (b) Volume reduction at the initial bubble escape and fluctuation amplitude of the bubble volume V_{bubble} as a function of the normalized effective path length for the gas transport L_{path} .

We have improved the theoretical model for the bubble escape dynamics by incorporating the effect of flow resistance that increases with the path length proportional to the thickness of cryogel overlayer. The reduction in V_{bubble} at the initial escape increased due to the λ_{pore} (Fig. 3e) but decreased by the effective L_{path} (Fig. R11b). Thus, the J_{ph} recovery of the cryogel-coated devices is determined by not only λ_{pore} but also effective L_{path} , which depend on the cryogel thickness. The fluctuation amplitude of V_{bubble} was decreased by both the λ_{pore} (Fig. 3e) and effective L_{path} (Fig. R11b).

In the revised manuscript, we modified the theoretical model and discussed the effect of the gas transport through the internal pore on the bubble dynamics. **Fig. R11** was included as **Supplementary Fig. 17** in the revised manuscript.

Revision made (colored in blue):

(in Page 12)

This result indicates that the large initial J_{ph} recovery followed by a relatively small J_{ph} fluctuation can be attributed to the local fracture caused by the stress localized in the overlayer. ~~As λ_{pore} increased, the reduction in V_{bubble} at the initial escape increased, while the fluctuation amplitude of V_{bubble} decreased (Fig. 3e).~~

For escaping of trapped bubbles, they should be transported through not only fractured pore but micropores present in the path for gas transport. We could observe the gas transport through the layer of micropores present between the trapped bubble and the surface of the cryogel overlayer (**Supplementary Fig. 17a**). At a given pressure difference in a channel, the gas transport rate is inversely proportional to the path length (L_{path}), that can increase in the thicker overlayer, according to Poiseuille's Law³⁵. Additionally, the resistance for gas transport can be increased by the elastic expansion and contraction of the pores. The distensibility of cryogel should be taken into account to estimate an effective value of L_{path} . The reduction in the V_{bubble} at the initial escape increased with λ_{pore} (**Fig. 3e**) but decreased with the effective L_{path} (**Supplementary Fig. 17b**). The fluctuation amplitude of the V_{bubble} was decreased by both λ_{pore} and effective L_{path} (**Fig. 3e** and **Supplementary Fig. 17b**). This result is consistent with the experimental result that the photocurrent fluctuation decreased with increasing overlayer thickness (**Supplementary Fig. 912**).

(in Page 26 of Supplementary Information)

The flow rate of the escaping air from the deflating rubber balloon can depend on the volume and pressure of the air and the radius of the inlet of balloon¹⁴. We assumed that ~~In this study~~, the volume of the bubble escaping from the pore V_{escape} is ~~assumed to be~~ a function of the volume and pressure of the bubble and the pore size. ~~as follows:~~

In addition to the fractured pore, the trapped gas bubble should pass through the internal micropores as observed in **Supplementary Fig. 17a**. The gas transport rate is decreased by the path length which is increased by the cryogel thickness due to the resistance by the internal pores of the cryogel according to Poiseuille' Law¹⁴. The elastic deformation of micropores can also increase the resistance for the gas transport. Therefore, the effective path length would be determined by not only the geometry of the pores but the elastic properties of the cryogel.

The V_{escape} is calculated as follows:

$$V_{\text{escape}} = (V_{\text{bubble,init}} \times A_0) \times \left(\frac{V_{\text{bubble}}}{V_{\text{bubble,init}}} \right)^\alpha \times \left(\frac{P_{\text{bubble}}}{P_{\text{bubble,init}}} \right)^\beta \times \left(\frac{\xi_{\text{bubble}}}{\xi_{\text{bubble,init}}} \right)^\delta \times \left(\frac{1}{L_{\text{path}}} \right)^\varepsilon \quad (10)$$

, where A_0 , α , β , δ , and ε are fitting parameters. L_{path} is a normalized effective path length.

Comment 1 (Some statements need to be checked and modified.):

Comment on (previously) Page 23: I believe the authors have an error in their description of the degree of J_{ph} drop. Instead of stating, “The degree of J_{ph} drop was determined as the photocurrent density at the time just before the occurrence of the photocurrent recovery divided by the initial photocurrent density,” it should be revised to: “The degree of J_{ph} drop is calculated as the difference between the initial photocurrent density and the photocurrent density right before the occurrence of the photocurrent recovery, all divided by the initial photocurrent density.”

Response:

We appreciate the reviewer’s critical comment regarding the statement of the calculation of the degree of J_{ph} drop. As the reviewer commented, the degree of J_{ph} drop, mentioned in our previous response to the comment 4 of the reviewer 3, was calculated as “the difference between the initial photocurrent density and the photocurrent density right before the occurrence of the photocurrent recovery, all divided by the initial photocurrent density.”

We just note that the description to calculate the J_{ph} drop was used to address the reviewer’s comment in the previous rebuttal and was not included in the main text.

Comment 2 (Some statements need to be checked and modified.):

Manuscript (Page 9, Line 2): The assertion, “We observed that the amplitude of the J_{ph} fluctuation was small when the initial J_{ph} drop recovery was large, and the J_{ph} degradation during the long-term operation was faster when the J_{ph} fluctuation was large,” seems to contradict the experimental findings presented. Specifically, for CG-1200 μm , both the initial J_{ph} drop recovery and the J_{ph} fluctuations are minimal (Fig. 1c).

Response:

We appreciate the reviewer’s comment regarding the phrase “ J_{ph} drop recovery.” We agree with the reviewer that “ J_{ph} drop recovery” could be confusing.

To eliminate any ambiguity related to the expression “ J_{ph} drop recovery,” we clearly defined “ J_{ph} drop” and “ J_{ph} recovery” as a change in J_{ph} at the initial drop and the following recovery, respectively, in the revised manuscript. We also indicated “ J_{ph} drop” and “ J_{ph} recovery” in the plot in **Fig. 1c** in the revised manuscript.

As shown in Supplementary Fig. 9 of the original manuscript, the J_{ph} fluctuation amplitude decreased by the overlayer thickness ranged from 400 μm to 1200 μm . Both the drop and recovery of the J_{ph} increased from the CG-400 μm to the CG-800 μm . However, the CG-1200 μm exhibited the similar J_{ph} drop or lower J_{ph} recovery compared to the CG-800 μm . Thus, we agree with the reviewer that the phrase “the J_{ph} fluctuation was small when the initial J_{ph} drop recovery was large” did not accurately describe the experimental data regarding the drop, recovery, and fluctuation amplitude of the J_{ph} .

In the revised manuscript, the phrase “the amplitude of the J_{ph} fluctuation was small when the initial J_{ph} drop recovery was large” was deleted. Regarding the effect of cryogel thickness, we added the following sentence “We found that both the fluctuation amplitude and degradation rate decreased with the overlayer thickness (**Supplementary Fig. 12**).”

Revision made (colored in blue):

(in Page 6)

Interestingly, after the ~~initial J_{ph} drop and~~ recovery, the ~~recovered~~ J_{ph} was maintained with small and regular fluctuations. *CG-800 μ m* and *CG-1200 μ m* exhibited a large initial J_{ph} drop ~~and~~ recovery within 0.5 h, followed by a ~~slight stable J_{ph} fluctuation value with slight fluctuations~~. The initial parabolic decreasing profile ~~during the initial J_{ph} drop~~ implies that the produced bubble continuously grows inside the cryogel, which would prevent the scattering of incoming light and reduce the mass transfer of electrolytes, thereby temporarily reducing the J_{ph} of devices.

(in Page 7)

On the other hand, the initial J_{ph} drop ~~recovery~~ was not observed for *CG-100 μ m* and *CG-200 μ m*, which exhibited relatively poor device lifetimes (**Supplementary Fig. 6a8a**). *CG-100 μ m* and *CG-200 μ m* exhibited significant J_{ph} degradation after 20 h of operation. Because the initial J_{ph} drop ~~recovery~~ was not observed and J_{ph} exhibited an irregular fluctuation, the time-dependent J_{ph} for thin overlayers was similar to that of the device without an overlayer.

(in Page 9)

We found that both the fluctuation amplitude and degradation rate decreased with the overlayer thickness ~~observed that the amplitude of the J_{ph} fluctuation was small when the initial J_{ph} drop recovery was large and the J_{ph} degradation during the long term operation was faster when the J_{ph} fluctuation was large~~ (**Supplementary Fig. 912**). Since the regular J_{ph} fluctuation was observed only after the drop and recovery of J_{ph} , the bubble dynamics including ~~This result indicates that bubble~~ expansion and escape ~~occurred~~ during the ~~initial J_{ph} drop and~~ recovery, respectively, can be critical in determining the photocurrent of the device with a cryogelated overlayer.

To determine the bubble dynamics, including nucleation, expansion, and escape during the initial J_{ph} drop ~~recovery~~ and the following ~~J_{ph} recovery and~~ fluctuation of J_{ph} , we visualized bubbles in the cryogelated overlayer *in situ* for *CG-800 μ m*, which showed a large initial J_{ph} drop ~~recovery~~ (**Fig. 2a**).

(in Page 10)

The size of the escaped bubbles was smaller in the repeated cycles than in the initial escape (**Fig. 2a**), which was consistent with the reduction in the amplitude of the J_{ph} fluctuation relative to the initial J_{ph} drop and the following recovery of J_{ph} .

(Figure 1)

Fig. 1. Cryogelated polyacrylamide overlayer for photoelectrochemical (PEC) devices. ... The inset graph in (c) indicates the schematic of the representative pattern of the J_{ph} -time profile of the PEC device with the cryogel overlayer during 0-1 h. ...

Comment 3 (Some statements need to be checked and modified.):

Manuscript (Page 2, Line 9): The statement, “Tests for cryogel overlayers with varied microstructures reveal that the hydrogen gas trapped in the cryogel protector reduce shear stress at the catalyst surface by providing bubble nucleation sites.” can be misinterpreted. The phrase “cryogel overlayers with varied microstructures” might be misconstrued as “cryogel overlayers with different pore structures.” However, throughout this paper, the pore structure of the cryogel overlayers remain consistent.

Response:

We appreciate the reviewer’s comment regarding the statement “cryogel overlayers with varied microstructures.” In **Figs. 1, 2, and 4**, the pore structure of the cryogel overlayers was interconnected micropores. However, we also conducted experiments with the PEC devices coated with cryogel overlayers with different pore structures, including ‘disconnected micropores’ and ‘surface micropores’ (**Figs. 5a and 5b of the original manuscript**).

Fig. 5. Effect of the microstructure of the cryogelated overlayer on the functional and structural stability of the photoelectrochemical (PEC) device. Schematic of the cross-sectional structure of the as-fabricated device with an overlayer, ... of the device surface of (a) *disconnected micropores* and (b) *surface macropores*. ...

In the revised manuscript, we clearly stated the types of different pore structures.

Revision made (colored in blue):

(in Page 2)

Tests for cryogel overlayers with varied ~~microstructures~~ pore structures, such as disconnected micropores, interconnected micropores, and surface macropores, reveal that the hydrogen gas trapped in the cryogel protector reduce shear stress at the catalyst surface by providing bubble nucleation sites.

REVIEWERS' COMMENTS

Reviewer #2 (Remarks to the Author):

The authors have now addressed all the questions, and the revised MS can be published without change.

Reviewer #4 (Remarks to the Author):

This paper was revised according to our comments. The sentence "However, solar water splitting systems suffer from short lifetimes due to catalyst instability, which might be attributed to the mechanical stress produced by hydrogen bubbles, preventing them from being used commercially." in the abstract is not appropriate and should be revised as the author also agrees that the degradation mechanism of a PEC system is not purely stress-initiated.

Point-by-point response to reviewers' comments

Title: Stable water splitting using photoelectrodes with a cryogelated overlayer

Author(s): Byungjun Kang[†], Jaiwan Tan[†], Kyungmin Kim, Donyoung Kang, Hyungsoo Lee, Sunihl Ma, Young Sun Park, Juwon Yun, Soobin Lee, Chan Uk Lee, Gyumin Jang, Jeongyoub Lee, Jooho Moon*, Hyungsuk Lee*

<Reviewer #2>

Remark:

The authors have now addressed all the questions, and the revised MS can be published without change.

Response:

We thank the reviewer for the recommendation of our revised manuscript for acceptance in Nature Communications. We sincerely appreciate the reviewers' time and efforts in evaluating our manuscript. All the critical and constructive comments provided by the reviewer significantly helped us to improve the quality of our manuscript.

<Reviewer #4>

Remark:

This paper was revised according to our comments. The sentence "However, solar water splitting systems suffer from short lifetimes due to catalyst instability, which might be attributed to the mechanical stress produced by hydrogen bubbles, preventing them from being used commercially." in the abstract is not appropriate and should be revised as the author also agrees that the degradation mechanism of a PEC system is not purely stress-initiated.

Response:

We thank to the reviewer for the dedicated time and effort in evaluating our manuscript. We could improve the quality of our manuscript due to the critical and beneficial comments made by the reviewer.

In the abstract of the revised manuscript, we modified the statement regarding the origin of the short lifetime of the solar water splitting systems.

Revision made (colored in blue):

(in Abstract)

However, solar water splitting systems suffer from short lifetimes due to catalyst instability, ~~which might be that~~ is attributed to both chemical dissolution and ~~the~~ mechanical stress produced by hydrogen bubbles, ~~preventing them from being used commercially~~.